# AMPED: Adaptive Multi-objective Projection for balancing Exploration and skill Diversification

**Geonwoo Cho[1,*]  Jaemoon Lee[2,*]  Jaegyun Im[1]  Subi Lee[1]  Jihwan Lee[1]  Sundong Kim[1]**
[1]Gwangju Institute of Science and Technology  [2]Seoul National University
{gwcho.public, dlwoans0001, jaegyun.public, soobi1234, qhddl2650, sdkim0211}@gmail.com

## Abstract

Skill-based reinforcement learning (SBRL) enables rapid adaptation in environments with sparse rewards by pretraining a skill-conditioned policy. Effective skill learning requires jointly maximizing both exploration and skill diversity. However, existing methods often face challenges in simultaneously optimizing for these two conflicting objectives. In this work, we propose a new method, Adaptive Multi-objective Projection for balancing Exploration and skill Diversification (AMPED), which explicitly addresses both: during pre-training, a gradient-surgery projection balances the exploration and diversity gradients, and during fine-tuning, a skill selector exploits the learned diversity by choosing skills suited to downstream tasks. Our approach achieves performance that surpasses SBRL baselines across various benchmarks. Through an extensive ablation study, we identify the role of each component and demonstrate that each element in AMPED is contributing to performance. We further provide theoretical and empirical evidence that, with a greedy skill selector, greater skill diversity reduces fine-tuning sample complexity. These results highlight the importance of explicitly harmonizing exploration and diversity and demonstrate the effectiveness of AMPED in enabling robust and generalizable skill learning. https://geonwoo.me/amped/

## 1 Introduction

Efficient exploration remains a major challenge in reinforcement learning (RL), particularly in environments with sparse or delayed rewards (Sutton & Barto, 2018; Schmidhuber, 2010; Vinyals et al., 2017; Litman, 2005). While biological agents naturally discover rewarding behaviors, artificial agents often rely on handcrafted reward functions, which demand extensive domain knowledge and limit scalability (Kwon et al., 2023). Skill-Based Reinforcement Learning (SBRL) addresses this by pretraining a skill-conditioned policy through unsupervised skill discovery (Gregor et al., 2016; Shi et al., 2022), enabling efficient adaptation to downstream tasks.

A common approach in SBRL is to use Unsupervised Reinforcement Learning (URL) objectives during pretraining to discover diverse and useful skills (Eysenbach et al., 2019; Gregor et al., 2016). Two widely used URL objectives are: (1) maximizing mutual information (MI) between skills and their state trajectories to promote skill diversity, and (2) maximizing state entropy to encourage exploration (Laskin et al., 2022; Liu & Abbeel, 2021b) (Appendix F). However, MI-driven objectives often induce premature specialization by curtailing exploration (Campos et al., 2020; Jiang et al., 2022; Strouse et al., 2022), while entropy-based exploration sacrifices skill distinguishability, limiting downstream utility. The core problem is how to balance these competing objectives, without resorting to ad-hoc heuristics.

In this work, we bridge two URL paradigms in the theoretical framework of multi-objective reinforcement learning, proposing Adaptive Multi-objective Projection for balancing Exploration and skill Diversification (AMPED). Few previous studies, such as CeSD (Bai et al., 2024) and ComSD (Liu et al., 2025), have explored similar integrations but either lack a solid theoretical foundation or exhibit significant limitations (Appendix G).

---

*Equal contribution.

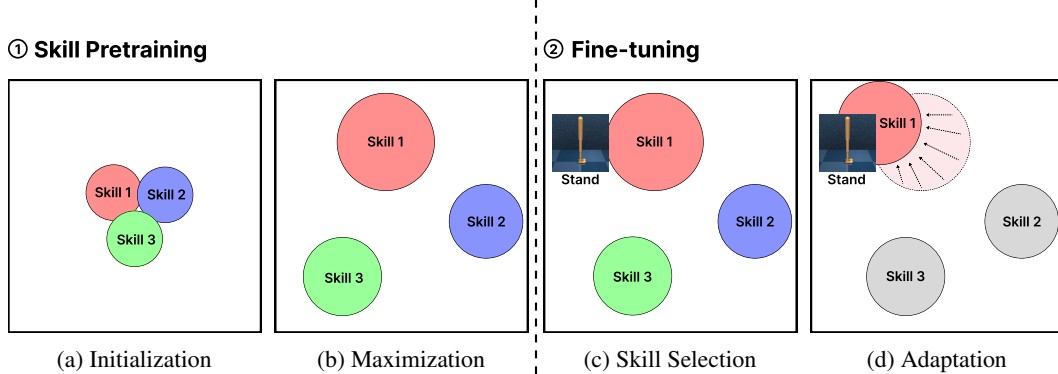

Figure 1: **Graphical scheme explaining our method, AMPED.** (a) At initialization, the skills exhibit small coverage that are close to each other in the task space. (b) During skill pretraining, exploration and diversity objectives encourage skills to widen and repel each regions. (c) In fine-tuning, the skill selector identifies the skill best aligned with the target task at each step. (d) The selected skill is further adapted via extrinsic rewards to maximize performance on the target task.

Our key insight is that gradient conflicts naturally arise between diversity objectives and exploration objectives, leading to inefficient updates that hinder learning (Yu et al., 2020). To address this issue, we adopt a gradient surgery method inspired by multi-objective RL, ensuring that conflicting gradient components are removed before applying updates (Yu et al., 2020). We use particle-based entropy and Random Network Distillation (RND) (Burda et al., 2019) to drive exploration, and adopt the AnInfoNCE objective for skill diversity. Furthermore, rather than selecting skills uniformly at random during fine-tuning, as is common in prior SBRL approaches (Bai et al., 2024; Eysenbach et al., 2019; Yang et al., 2023), we introduce a Soft Actor-Critic (SAC) based skill selector that learns to select the best matching pretrained skill (Haarnoja et al., 2018). This adaptive selection manner maximally leverages the inherent diversity of the skill repertoire. A graphical summary of these contributions is provided in Figure 1.

AMPED improves performance over strong baselines, DIAYN, COMSD, RND, CeSD, BeCL, CIC, and APT, across benchmarks. We evaluate AMPED on Maze environments (Campos, 2020) and the Unsupervised Reinforcement Learning Benchmark (URLB) (Laskin et al., 2021). In the Maze suite, AMPED learns well-separated skills while simultaneously achieving high state coverage, whereas competing methods fail to maximize both. On URLB, AMPED delivers statistically significant improvements in return over the baselines. Taken together, these results show that explicitly resolving exploration-diversity gradient conflicts yields substantial gains in SBRL.

Ablation studies further confirm that each component of our framework, entropy bonuses, RND, Anisotropic InfoNCE, gradient surgery, and the skill selector, contributes meaningfully to overall performance. Moreover, we show, theoretically and empirically, that greater skill diversity reduces fine-tuning sample complexity when paired with a greedy skill selector. This clarifies the respective roles of diversity and selection and motivates further work on principled skill selector design.

We provide the full implementation of AMPED to facilitate the reproduction of our main results at https://github.com/Cho-Geonwoo/amped.

## 2 PRELIMINARIES

### 2.1 MARKOV DECISION PROCESS (MDP) AND CONDITIONAL MDP (CMDP)

MDP is a tuple $\mathcal{M} := (\mathcal{S}, \mathcal{A}, P, R, \mu, \gamma)$, where $\mathcal{S}$ is a state space; $\mathcal{A}$ is an action space; $P : \mathcal{S} \times \mathcal{A} \to \Delta(\mathcal{S})$ is a transition model where we denote the probability of transitioning from $s$ to $s'$ with action $a$ by $P(s'|a, s)$; $R : \mathcal{S} \times \mathcal{A} \to \mathbb{R}$ is a reward function; $\mu \in \Delta(\mathcal{S})$ is the initial state distribution; and $\gamma \in (0, 1]$ is a discount factor. A trajectory is a sequence of states and actions, for example: $\tau = (s_1, a_1, s_2, a_2, \ldots a_{H-1}, s_H)$. We will only consider finite horizon MDP, i.e. $H < \infty$. A policy $\pi : \mathcal{S} \to \Delta(\mathcal{A})$ maps states to action probabilities, denoted as $\pi(a|s)$.

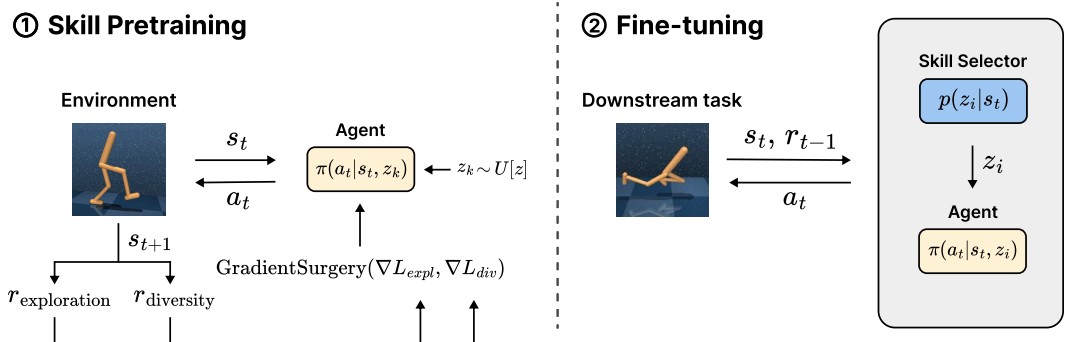

Figure 2: **Overview of the training process of AMPED.** During the skill pretraining phase, the agent is conditioned on randomly sampled skills and optimized using intrinsic rewards for exploration and diversity. These gradients are not directly used, but are balanced via a gradient surgery mechanism. In fine-tuning phase, a skill selector adaptively selects skills on each step, based on task-specific feedback, and the agent is further optimized using extrinsic rewards from the downstream target task.

CMDP extends MDP by introducing a latent variable $z \in \mathcal{Z}$, often representing a skill or context. The policy becomes $\pi(a|s, z)$, additionally conditioned on $z$. CMDP is used in skill discovery, where the goal is to learn diverse, distinct behaviors parameterized by $z$.

Both MDP and CMDP aim to maximize the expected cumulative discounted reward: $\max_\pi \mathbb{E}_{\tau \sim (\pi, P)}[\sum_t \gamma(t) r_t]$ where trajectory $\tau$ is generated from policy $\pi$.

## 2.2 ENTROPY AND MUTUAL INFORMATION

For random variables $X, Y$, Shannon entropy and MI are defined as $H(X)$, $I(X; Y)$, respectively:

$$H(X) = -E[\log p(X)], \quad I(X; Y) = D_{\mathrm{KL}}\big(p_{X,Y} \,\|\, p_X\, p_Y\big) \tag{1}$$

where $D_{\mathrm{KL}}$ is a Kullback-Leibler divergence, and $p_{X,Y}$ is a joint distribution. Higher entropy corresponds to higher unpredictability, the state distribution becomes more uniform in the state space, thereby facilitating broader exploration. In contrast, higher MI indicates stronger statistical dependence between two random variables. MI is commonly used between skills and trajectories, so that each skill reliably produces its characteristic behavior. Moreover, by using contrastive learning to estimate MI, each skill repel the others, thereby achieving skill diversity.

These information-theoretic terms are widely used as an intrinsic objective in URL. For example, CIC (Laskin et al., 2022): $I(\tau; z)$, DIAYN (Eysenbach et al., 2019): $I(S; Z) + H(A|S) - I(A; Z|S)$, BeCL (Yang et al., 2023): $I(S^{(1)}; S^{(2)})$. Details on the objectives are provided in Appendix H.

## 2.3 GRADIENT CONFLICT

In multi-objective RL, optimizing multiple objectives simultaneously with the same network can lead to conflicts between the gradients of each objective. A naive implementation computes the gradients for each objective independently and performs gradient descent using their sum. However, this can result in gradient conflict, where the update direction that benefits one task negatively impacts another (Yu et al., 2020).

To address this issue, Yu et al. (2020) proposed PCGrad, a gradient surgery method designed to mitigate such conflicts by removing interfering gradient components. Given a set of objectives $\mathcal{L}_k(\theta)$ for $k = 1, \ldots, n$, the corresponding gradients are first computed as $g_k = \nabla_\theta \mathcal{L}_k$. The gradients are then processed sequentially in a random order. For each pair of gradients $g_i$ and $g_k$, if a conflict is detected, i.e., if $g_i \cdot g_k < 0$, then the projection of $g_i$ onto $g_k$ is subtracted from $g_i$. Thus, the modified gradient is guaranteed not to interfere with the descent directions of other tasks. Moreover, PCGrad paper showed that under appropriate conditions, a projected gradient step can outperform standard stochastic gradient descent (SGD). Finally, the adjusted gradients are aggregated and applied using conventional SGD.

# 3 ADAPTIVE MULTI-OBJECTIVE PROJECTION FOR EXPLORATION AND DIVERSIFICATION (AMPED)

Our goal is to maximize both skill diversity and exploration, as supported by prior works (Eysenbach et al., 2019; Gregor et al., 2016; Yang et al., 2023; 2024). Previous methods in URL, such as CeSD and ComSD (Bai et al., 2024; Liu et al., 2025), have combined these objectives. Following this approach, we optimize the discounted cumulative return $\mathbb{E}\left[\sum_t \gamma^t (r_{\text{exploration}} + r_{\text{diversity}})\right]$, using a DDPG agent (Lillicrap et al., 2015). Here, $r_{\text{exploration}}$ incorporates entropy and RND-based objectives (Burda et al., 2019), while $r_{\text{diversity}}$ includes the AnInfoNCE term (Rusak et al., 2024). The specific formulations and the rationale for their use are detailed in Section 3.2. We illustrate our overall method in Figure 2.

Maximizing state entropy is essential because it induces a uniform visitation distribution, minimizing worst-case regret as shown by Gupta et al. (2018). And this principle has been empirically validated in prior works (Jain et al., 2023; Liu & Abbeel, 2021b). We now briefly motivate the importance of skill diversity for downstream tasks via the following theoretical analysis.

## 3.1 THEORETICAL ANALYSIS OF SKILL DIVERSITY

Assume a finite state space $\mathcal{S}$ with cardinality $S$, and a finite horizon $H$. Suppose we are given skill-conditioned policies $\pi(a \mid s, z)$ with a finite number of skills and a downstream task. Let $\pi$ be the target policy and $\rho \in \Delta(\mathcal{S})$ be a corresponding discounted state occupancy measure. Also set $z_\star$ as a best policy in the sense that $z_\star = \arg\min_z d(\rho, \rho_z)$. We denoted the total variation of two probability distributions by $d(\rho_1, \rho_2) = \frac{1}{2}\|\rho_1 - \rho_2\|_1$.

**Theorem 1.** *Define* $\delta = \min_{i \neq j} d(\rho_{z_i}, \rho_{z_j})$, $\varepsilon = d(\rho, \rho_{z_\star})$. *Assume that the skills are sufficiently diversified, so that* $\Delta \equiv \delta - 2\varepsilon > 0$.

*Draw* $n$ *i.i.d. trajectories from target policy* $S^{(1)}, \ldots, S^{(n)} \sim \pi$ *and form the empirical state distribution* $\widehat{\rho}$. *Consider the greedy skill selector* $\widehat{z} := \arg\min_z d(\widehat{\rho}, \rho_z)$. *Then*

$$\Pr[\widehat{z} \neq z_\star] \leq 2^S H \exp\left(-\frac{n\Delta^2}{2}\right). \tag{2}$$

*In terms of confidence level* $\eta \in (0, 1)$, *if* $n \geq \frac{2}{\Delta^2}(S \log 2 + \log H - \log \eta)$, *we have* $\Pr[\widehat{z} \neq z_\star] \leq \eta$.

Thus, greater diversity between skills ($\delta$) increases the margin $\Delta$ and reduces the number of required samples to skill selector be optimal. This formalizes the intuition that diverse skills ease identifying the skill whose induced policy is closest to the target policy $\pi$. Proof of this theorem is provided in Appendix A.

## 3.2 EXPLORATION & DIVERSITY INTRINSIC REWARDS

**Exploration Reward.** Our exploration reward consists of two components: an entropy-based term and a RND term (Burda et al., 2019). The entropy component, widely used in SBRL (Gregor et al., 2016; Laskin et al., 2022; Liu & Abbeel, 2021a), enhances exploration when maximized and is defined as $H(S_{\text{tot}})$, where $S_{\text{tot}}(s) = (1 - \gamma) \sum_t \gamma^t p(s_t = s)$ is the discounted state occupancy measure. Since the exact discounted state occupancy measure is unknown, we approximate it using a particle-based method by Laskin et al. (2022). Each particle is an embedded state pair $x_i = g_{\psi_1}(\tau_i)$ where $\tau_i = (s_t, s_{t+1})$ and $g_{\psi_1}$ denotes the embedding function. Then the distribution is estimated using distances to its $k$th nearest neighbor, $R_{i,k,n}$. The intrinsic reward is then computed as $r_{\text{entropy}}(s) = \log(\sum_{l=1}^{k} R_{i,l,n})$, which captures the entropy contribution of each particle.

To construct a meaningful latent space, we train an encoder with the contrastive loss:

$$\mathcal{L}_{\text{CIC}}(\tau) = \frac{g_{\psi_1}(\tau_i)^\top g_{\psi_2}(z_i)}{\|g_{\psi_1}(\tau_i)\|\|g_{\psi_2}(z_i)\|T} - \log \frac{1}{N} \sum_{j=1}^{N} \exp\left(\frac{g_{\psi_1}(\tau_j)^\top g_{\psi_2}(z_i)}{\|g_{\psi_1}(\tau_j)\|\|g_{\psi_2}(z_i)\|T}\right), \tag{3}$$

here $g_{\psi_2}$ encodes skills, and $T > 0$ is a temperature hyperparameter from CIC (Laskin et al., 2022).

Despite its benefits, entropy-based exploration alone is insufficient in high-dimensional spaces. The particle-based entropy estimator has a time complexity of $\mathcal{O}(n \log n)$, where $n$ denotes the number

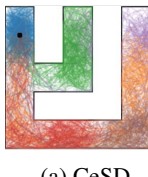 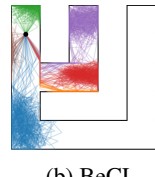

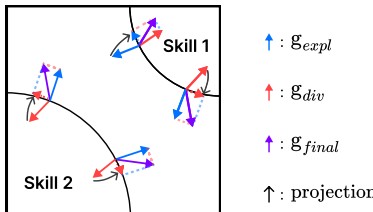

(a) CeSD      (b) BeCL

Figure 3: **Exploration trajectories in the Square Maze with six skills.** CeSD yields more contiguous coverage, while BeCL enforces stronger separation, leaving noticeable gaps.

Figure 4: **Graphical illustration of gradient surgery.** When diversity gradient (red) and exploration gradient (blue) conflict, one gradient is randomly projected onto the orthogonal complement of the other to balance updates. Added gradient (purple) is used for update of parameters.

of states in the replay buffer, because it requires computing the $k$-nearest neighbors for every state pair. This makes it impractical for large replay buffers. In practice, following Liu & Abbeel (2021a), we truncate the buffer and compute entropy using only the most recent 5,000 states, which reduces computational overhead but inevitably degrades the fidelity of the entropy estimate.

To address this limitation, we combine entropy bonuses with RND. RND trains a predictor network $f_\theta$ to match the output of a fixed, randomly initialized target network $f_{\text{target}}$, producing an intrinsic reward $r_{\text{rnd}}(s) = \|f_\theta(s) - f_{\text{target}}(s)\|^2$, where larger prediction error indicates novel states and thus encourages exploration. Importantly, RND provides a model-based intrinsic reward with linear complexity in $n$ (i.e., $\mathcal{O}(n)$). However, when data are scarce in early training, RND can be noisy and unstable.

By leveraging entropy-based exploration to supply a reliable signal when the replay buffer is small, and using RND to compensate for the computational and statistical limitations of the particle-based estimator as the buffer grows, the two methods complement each other and together provide a more robust exploration mechanism.

We define the exploration reward as a linear combination of the RND and entropy terms. Specifically, $r_{\text{exploration}}(s) = \alpha r_{\text{entropy}}(s) + \beta r_{\text{rnd}}(s)$ where $\alpha$ and $\beta$ is the positive scaling coefficients that modulate the relative influence of the entropy-based and RND rewards. Ablation studies in Appendix D confirm that combining entropy and RND significantly improves exploration efficiency in high-dimensional environments.

**Diversity Reward.** To motivate our diversity reward formulation, we first revisit CeSD (Bai et al., 2024). CeSD optimizes $H(S_{\text{tot}}) + \mathcal{L}_{\text{diversity}}$, where $\mathcal{L}_{\text{diversity}}$ encourages trajectories generated by different skills to avoid overlapping.

However, this term treats all non-overlapping trajectories equally and therefore fails to distinguish between skills that are only mildly separated and those that are strongly separated (see Appendix G.1 for details). To address this limitation, we adopt the mutual information objective $I(S^{(1)}, S^{(2)})$ from BeCL (Yang et al., 2023), where $S^{(1)}$ and $S^{(2)}$ denote states sampled from trajectories induced by the same skill $z$. Maximizing $I(S^{(1)}, S^{(2)})$ encourages states generated by the same skill to cluster together while pushing apart those generated by different skills.

Unlike CeSD's heuristic penalty, the MI-based formulation explicitly pushes apart skill distributions, resulting in substantially stronger skill separation. Furthermore, BeCL shows that sufficiently maximizing this objective also increases state entropy, thereby jointly promoting exploration and diversity. Our 2D maze experiment (Figure 3) empirically confirms that the MI objective produces clearer skill differentiation than CeSD. A detailed discussion of skill diversification under the BeCL objective is provided in Appendix B.

For MI estimation, we employ AnInfoNCE (Rusak et al., 2024), an anisotropic variant of InfoNCE (Oord et al., 2018) designed to capture asymmetries in latent factors. To the best of our knowledge, AnInfoNCE has not previously been used for skill diversification. However, our empirical results (Section 4.3) show that it provides a more effective MI estimate than standard InfoNCE in this setting, motivating its use in AMPED.

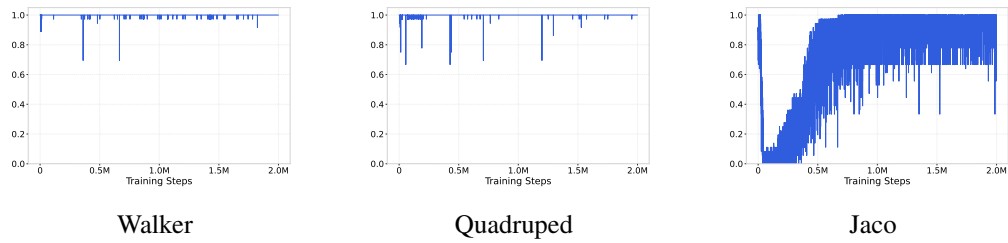

Figure 5: **Evolution of gradient conflict ratio in URLB environment.**

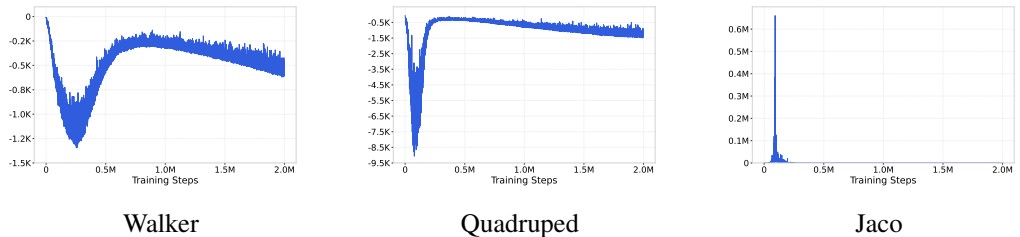

Figure 6: **Evolution of gradient similarity in URLB environment.**

The AnInfoNCE loss is defined as:

$$\mathcal{L}_{\text{AINCE}}(f, \hat{\Lambda}) = -\mathbb{E}_{s,s^+,\{s_i^-\}} \left[ \ln \frac{e^{-\|f(s^+)-f(s)\|_{\hat{\Lambda}}^2}}{e^{-\|f(s^+)-f(s)\|_{\hat{\Lambda}}^2} + \sum_{i=1}^{M} e^{-\|f(s_i^-)-f(s)\|_{\hat{\Lambda}}^2}} \right], \tag{4}$$

where $s, s^+$ are positive samples (from the same skill), and $\{s^-\}$ are negative samples (from different skills). The matrix $\hat{\Lambda}$ is a learnable diagonal matrix, and the weighted norm is defined as $\|x\|_{\hat{\Lambda}}^2 = x^T \hat{\Lambda} x$. The state encoder $f$ and $\hat{\Lambda}$ are updated via loss minimization. Accordingly, we define contrastive reward $r_{\text{diversity}}$ as the term inside the bracket.

## 3.3 BALANCING EXPLORATION AND DIVERSITY OBJECTIVES

Differentiable skills improve adaptability in dynamic skill selection but often compromise exploration ability (Yang et al., 2023), leading to suboptimal performance in environments. By treating diversity and exploration as distinct objectives, our problem can be framed as a multi-objective RL setting, in which two objectives are optimized concurrently. In this perspective, observed gradient interference can be interpreted as a form of *gradient conflict*, a well-documented challenge in multi-objective RL.

Figure 5 reports the *gradient conflict ratio* throughout training, averaged over three seeds. The conflict ratio at each step denotes the fraction of minibatch samples whose exploration and diversity gradients exhibit negative inner products: $\langle g_{\text{expl}}, g_{\text{div}} \rangle < 0$. Across all URLB domains, we observe substantial gradient conflict (Walker: $0.9997 \pm 0.0004$, Quadruped: $0.9997 \pm 0.0006$, Jaco: $0.7777 \pm 0.1256$), indicating persistent disagreement between objectives. To further characterize the nature of these interactions, Figure 6 visualizes the *gradient similarity* (raw inner product). These results underscore the importance of explicitly resolving gradient conflicts to enable stable optimization when combining exploration and diversity objectives.

To mitigate this issue, we integrate a gradient projection method, known as gradient surgery or projecting conflicting gradients (PCGrad), proposed by Yu et al. (2020). The key idea is to remove gradient interference by projecting one objective's gradient onto the orthogonal complement of the other (Figure 4). At each update, we randomly choose which gradient to adjust: with probability $p$ we project $g_{expl}$ to $g_{div}$, and with probability $1 - p$ vice versa. Then, the final update gradient $g_{final}$ is obtained by summing two gradients, one projected. The procedure is detailed in Algorithm 1. Although more advanced methods exist (e.g., Liu et al. (2021); Navon et al. (2022)), we opted for the original gradient surgery approach due to its simplicity and ease of integration. Despite its straightforward design, this method proved sufficiently effective in mitigating gradient conflicts for our application.

### 3.4 ADAPTIVE SKILL SELECTION

To fully exploit the diversity of the learned skill set, we adopt a skill-selection mechanism during downstream fine-tuning. Specifically, we train a skill selector $p(z \mid s)$ jointly with the policy. At each time step, the selector samples a skill according to $p(z \mid s)$, and the policy conditioned on the selected skill is updated using the downstream task reward. The skill selector is optimized to maximize the same downstream reward signal as the policy. Both the policy and the skill selector are updated at every training step. To balance exploration of new skills with exploitation of high-performing ones, we employ an $\epsilon$-greedy strategy with $\epsilon$ decaying over the course of training.

Prior methods often impose constraints to stabilize skill learning. For example, DIAYN (Eysenbach et al., 2019) freezes the prior distribution, VIC (Gregor et al., 2016) fixes the skill at initialization, and other approaches rely on labeled demonstrations. In contrast, our method jointly trains the policy and skill distribution, which turns out to be stable and effective.

During evaluation, the skill selector becomes deterministic, employing a greedy strategy to maximize task performance. This hierarchical framework facilitates efficient skill transfer and adaptation while maintaining decision stability. Detailed descriptions of the implementation of the skill selector are provided in Appendix I.4.

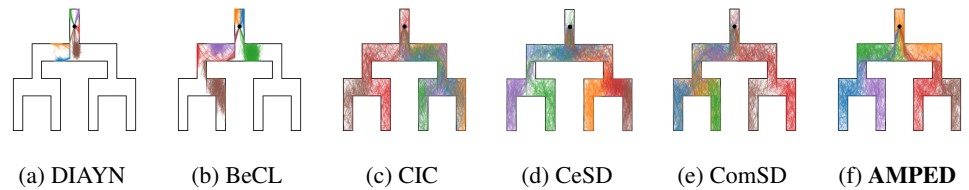

|          |          |         |          |           |            |
|----------|----------|---------|----------|-----------|------------|
| (a) DIAYN | (b) BeCL | (c) CIC | (d) CeSD | (e) ComSD | (f) **AMPED** |

Figure 7: **Agents exploring on Tree Maze after pretrained from different skill discovery objectives.** From (a) to (f) each are trained with six skills. Visually, our approach exhibits the most distinct skills while ensuring full coverage of the state space.

## 4 EXPERIMENTS

In this section, we report results on the URLB benchmark, which we use as our main quantitative evaluation of downstream performance and sample efficiency. Before that, we additionally present visualizations on the Tree Maze environment to provide qualitative insight into how our method explores the state space and separates skills. We then run a series of targeted ablations to study the impact of each algorithmic component (RND, AnInfoNCE, gradient surgery, and the skill selector), and conclude by visualizing representative skills learned during URLB pretraining.

### 4.1 SKILL DISCOVERY IN TREE MAZE

The experiment demonstrating the skill discovery capability is conducted in a Tree Maze environment (Campos et al., 2020). For details on the environment, implementations, and hyperparameters, refer to the Appendix I. The Tree Maze serves as a toy environment for preliminary analysis and insight; accordingly, we evaluate a reduced set of baselines, DIAYN, BeCL, CIC, CeSD, and ComSD, compared to those used in URLB. Refer to the Appendix H for comprehensive details on the baselines.

Figure 7 illustrates the visualization of each baseline's performance after pretraining with six skills. In terms of skill distinguishability, our observations indicate that DIAYN, BeCL are capable of learning distinct skills, enabling clear differentiation among the states covered by each skill. Conversely, the skills learned by CIC are less distinguishable, likely due to the absence of a skill-differentiating term in its reward function. Regarding state coverage, CIC, CeSD and ComSD nearly reach the state coverage limit, whereas DIAYN and BeCL exhibit inferior performance in this regard. Notably, our proposed method, AMPED, demonstrates superior performance in both maximizing skill discriminability and state coverage, achieving the state coverage limit while each skill clearly separated. Additional experiments on the effect of varying the number of skills, results in other maze layout, and the evolution of skills over training steps are provided in Appendix C.

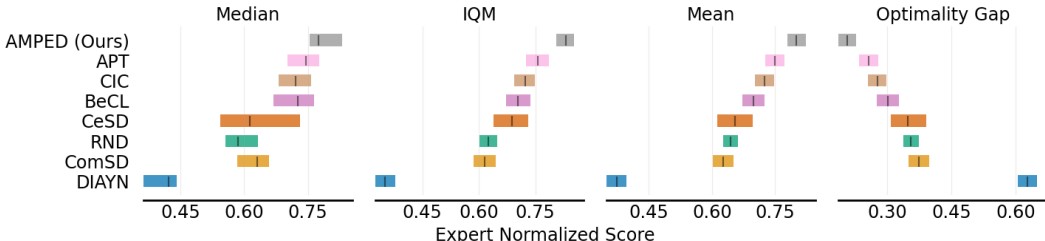

Figure 8: **Aggregated expert-normalized performance on 12 URLB downstream tasks.** Four metrics, median, IQM, arithmetic mean, and optimality gap, are plotted using the evaluation protocol introduced by Agarwal et al. (2021). Our method (gray) achieves the highest median, IQM, and mean scores and the smallest optimality gap, outperforming the previous state-of-the-art APT (pink) and other baselines.

## 4.2 EVALUATION ON URLB

To evaluate the performance of our method on downstream tasks, we utilize 12 tasks from the URLB. The benchmark comprises three domains: Walker, Quadruped, and Jaco. Detailed descriptions of the URLB are provided in Appendix I.3. Each method is first pretrained for 2M steps using an intrinsic reward, followed by fine-tuning for 100K steps on the downstream tasks.

For comparative evaluation, we selected strong baseline methods from the URLB, including DIAYN, APT, BeCL, CIC, RND, and the recently proposed CeSD and ComSD. Furthermore, methods such as LSD (Park et al., 2022), CSD (Park et al., 2023), and Metra (Park et al., 2024) were omitted because they do not exhibit performance improvements on the URLB relative to CeSD (Bai et al., 2024). Our implementation adheres to the official URLB code (Laskin & Yarats, 2025). Additional information on hyperparameters and network architectures can be found in Appendix I. For more details on reproducing these baselines, see Appendix I.8.

To ensure a fair comparison in Figure 8, we fine-tuned all methods under identical conditions and did not use the skill selector. Instead, following the downstream fine-tuning protocol of Yang et al. (2023), the skill was uniformly sampled from the full skill set and switched every 50 steps.

Unless otherwise stated, all experiments on URLB were pre-trained across 10 random seeds, and each fine-tuning run reused its corresponding pre-training seed. We aggregate statistics using the rliable RL framework (Agarwal et al., 2021), reporting median, mean, interquartile mean (IQM), and optimality gap. Here, the IQM is the average over the middle 50% of returns (between the 25th and 75th percentiles), and the optimality gap is the normalized difference between an expert reference score and the agent's score, as defined in (Agarwal et al., 2021). The expert score is derived from an expert DDPG agent following (Agarwal et al., 2021).

As shown in Figure 8, our method achieves the best results on the URLB. As recommended by Agarwal et al. (2021), we use the IQM as our primary performance measure. In particular, it surpasses the skill-differentiating methods BeCL by 17.96%, the entropy-maximization method CIC and APT by 15.02%, 9.73%, as well as the recent diversity-exploration hybrids CeSD and ComSD by 20.91%, 35.01%. These results suggest that considering both diversity and exploration is critical for downstream task performance, and more importantly, appropriately balancing these objectives is essential. All per-task scores for each method are reported in Appendix N.1. In addition, we include the evolution of the four evaluation metrics over training time for the Quadruped environment.

## 4.3 ABLATION STUDIES

In this section, we present the main analyses of AMPED. Additional results, including URLB reward-scaling ablations (Appendix D) and an extended evaluation of the skill-selection mechanism (Appendix E), further support our findings.

**Q1. Do all components of AMPED contribute to its performance?**

Table 1: **Episode returns under component ablation.** Ablating any single component, RND, AnInfoNCE loss, gradient surgery, or the skill selector, occasionally improves performance on individual tasks, yet yields degraded overall returns. **AMPED (Ours)** indicate the procedure including all of them; RND reward, AnInfoNCE, gradient surgery, and skill selector. The best result is shown in **bold**, and the second-best is underlined.

| Domain | Task | **AMPED (Ours)** | w.o. RND | w.o. AnInfoNCE | w.o. Gradient Surgery | w.o. Skill Selector |
|---|---|---|---|---|---|---|
| Walker | Flip | 674 $\pm$ 105 | 487 $\pm$ 47 | 536 $\pm$ 75 | 625 $\pm$ 48 | **686** $\pm$ 133 |
|  | Run | 467 $\pm$ 103 | 341 $\pm$ 67 | 440 $\pm$ 41 | 427 $\pm$ 57 | **517** $\pm$ 49 |
|  | Stand | **951** $\pm$ 38 | 917 $\pm$ 67 | 950 $\pm$ 25 | 939 $\pm$ 26 | 947 $\pm$ 19 |
|  | Walk | **929** $\pm$ 19 | 638 $\pm$ 60 | 923 $\pm$ 18 | 899 $\pm$ 45 | 886 $\pm$ 63 |
|  | Sum | 3021 | 2383 | 2849 | 2890 | **3036** |
| Quadruped | Jump | **720** $\pm$ 32 | 597 $\pm$ 154 | 705 $\pm$ 22 | 641 $\pm$ 64 | 699 $\pm$ 68 |
|  | Run | 494 $\pm$ 53 | 410 $\pm$ 84 | **496** $\pm$ 37 | 453 $\pm$ 13 | 493 $\pm$ 54 |
|  | Stand | **906** $\pm$ 67 | 905 $\pm$ 10 | 867 $\pm$ 70 | 890 $\pm$ 34 | 816 $\pm$ 150 |
|  | Walk | **890** $\pm$ 59 | 611 $\pm$ 228 | 870 $\pm$ 26 | 747 $\pm$ 114 | 816 $\pm$ 116 |
|  | Sum | **3010** | 2523 | 2938 | 2731 | 2824 |
| Jaco | Re. bottom left | 143 $\pm$ 32 | **147** $\pm$ 14 | 105 $\pm$ 33 | 111 $\pm$ 27 | 139 $\pm$ 34 |
|  | Re. bottom right | 144 $\pm$ 25 | 132 $\pm$ 40 | **148** $\pm$ 14 | 114 $\pm$ 35 | 140 $\pm$ 21 |
|  | Re. top left | 140 $\pm$ 39 | **163** $\pm$ 36 | 140 $\pm$ 23 | 96 $\pm$ 23 | 130 $\pm$ 38 |
|  | Re. top right | **154** $\pm$ 46 | 144 $\pm$ 47 | 92 $\pm$ 24 | 106 $\pm$ 49 | 146 $\pm$ 49 |
|  | Sum | 581 | **586** | 485 | 427 | 555 |

Ablation of AMPED shows that each component, RND, contrastive diversity, gradient surgery, and skill selection, makes a non-redundant and substantial contribution to the overall efficacy of AMPED. To quantify each component's impact, we individually ablated it within AMPED and reporting the resulting relative change in total returns with standard deviation (Table 1). In the Walker domain, removing RND incurs a 21.1% drop; in Quadruped it costs 16.2%; in Jaco, 0.9% increased but it is negligible, confirming RND's crucial exploration role. Dropping the AnInfoNCE diversity term reduces Walker by 5.7%, Quadruped by 2.4%, and Jaco by 16.5%, underscoring the need for skill separation. Disabling gradient surgery degrades returns by 4.3% (Walker), 9.3% (Quadruped), and 26.5% (Jaco), highlighting the value of conflict resolution. Finally, omitting the skill selector yields a 0.5% gain in Walker but decreases Quadruped by 6.2% and Jaco by 4.5%, demonstrating the importance of the skill selector.

**Q2. How does the gradient projection ratio affect downstream task performance?**

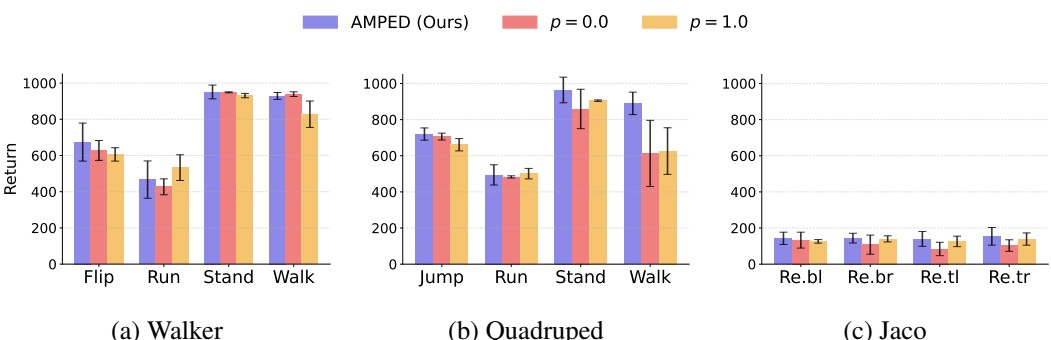

|  (a) Walker |  (b) Quadruped |  (c) Jaco |

Figure 9: **Task-level return comparison under different projection ratios.** In the Jaco domain, task labels "Re.bl", "Re.br", "Re.tl", and "Re.tr" refer to reaching the bottom-left, bottom-right, top-left, and top-right targets, respectively.

A balanced projection ratio effectively mitigates gradient conflicts and consistently improves skill learning across diverse environments (Figure 9). We compare three projection strategies: projecting

the exploration gradient onto the diversity gradient ($p = 0.0$), projecting the diversity gradient onto the exploration gradient ($p = 1.0$), and our default AMPED configuration, which uses a fixed, environment-specific projection ratio. The exact values of the projection ratios are provided in Appendix I.5. Results are reported as mean $\pm$ standard deviation; the $p = 0.0$ and $p = 1.0$ variants are averaged over three random seeds. AMPED achieves the highest aggregate returns in all three domains. Full numerical results for the projection ratio ablations are provided in Appendix N.2.

**Q3. How does the number of skills affect downstream task performance?**

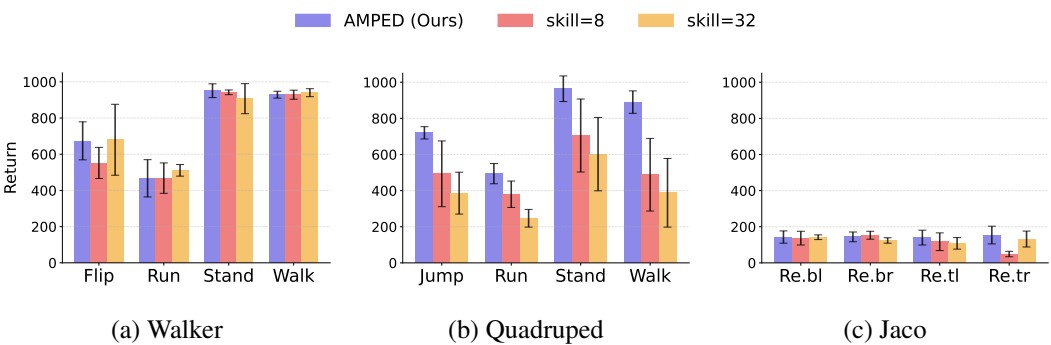

(a) Walker  (b) Quadruped  (c) Jaco

Figure 10: **Task-level return comparison under different numbers of skills.**

Increasing the number of skills does not necessarily increase diversity. As illustrated in Figure 12, a small skill set may already achieve complete coverage of the environment, leaving limited room for additional skills to expand diversity. Moreover, even in larger environments, diversity is optimized through an approximate mutual-information objective rather than an exact estimator, meaning that increasing the number of skills does not guarantee proportional increases in diversity. Consequently, choosing an appropriate skill dimension is critical for maximizing diversity and improving downstream performance. To examine this effect, we conduct an ablation study on the skill dimension.

We vary the skill dimension while keeping all other components of AMPED fixed and report the downstream performance in Figure 10. Our default configuration (16 skills) achieves the strongest results in Quadruped and Jaco, while performing competitively with the 32-skill variant in Walker. These findings indicate that an appropriately chosen skill dimension is essential. In our experiments, we follow prior work (Yang et al., 2023) and adopt 16 skills, but developing principled methods for selecting the optimal number of skills remains an interesting direction for future research. Full numerical results for the projection ratio ablations are provided in Appendix N.3.

## 5  CONCLUSION

In this work, we introduce AMPED to jointly address exploration and skill diversity in SBRL. Our framework unifies entropy-based exploration with contrastive skill separation, explicitly mitigates their gradient conflicts via PCGrad, and employs a skill selector to adaptively deploy skills during fine-tuning. Empirically, we find that (i) reducing exploration-diversity gradient interference leads to improved performance, (ii) combining AnInfoNCE-inspired diversity losses with RND-driven entropy bonuses balances the objectives. Taken together, our theoretical analysis highlights the crucial role of skill diversity in reducing sample complexity and enabling more effective skill selection.

While AMPED was developed for skill-based RL, its core insight, treating exploration and diversity as competing objectives and resolving their gradient conflicts via projection, extends to other settings with multiple learning signals, motivating broader use of gradient-projection methods. Future research could adopt more advanced conflict-resolution techniques and remove remaining heuristics, or identify alternative objective functions that better reconcile exploration and diversity. Further details are available in Appendix K. By tackling these challenges, the SBRL community can progress toward creating richer, more capable agents.

## ACKNOWLEDGMENTS

This work was supported by IITP (RS-2024-00445087; 10%, RS-2025-25410841; 10%, No. 2019-0-01842; 10%), NRF (RS-2024-00451162; 20%, RS-2025-02633598; 20%, RS-2025-25419920; 20%), GIST (KH0870; 10%) funded by the Ministry of Science and ICT, Korea. Computing resource were supported by GIST SCENT-AI, AICA, KISTI, and NIPA.

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

## APPENDIX

## A  PROOF OF THE THEOREM

Assume a finite state space $\mathcal{S}$ with cardinality $S$, and a finite horizon $H$. Suppose we are given skill-conditioned policies $\pi(a \mid s, z)$ with a finite number of skills and a downstream task. Let $\pi$ be the target policy and $\rho \in \Delta(\mathcal{S})$ be a corresponding discounted state occupancy measure. Also set $z_\star$ as a best policy in the sense that $z_\star = \operatorname{argmin}_z d(\rho, \rho_z)$. We denoted the total variation of two probability distributions by $d(\rho_1, \rho_2) = \frac{1}{2}\|\rho_1 - \rho_2\|_1$.

**Theorem 1.** *Define $\delta = \min_{i \neq j} d(\rho_{z_i}, \rho_{z_j})$, $\varepsilon = d(\rho, \rho_{z_\star})$. Assume that skills are diversified enough, so that $\Delta \equiv \delta - 2\varepsilon > 0$.*

*Draw $n$ i.i.d. trajectories from the target policy $S^{(1)}, \ldots, S^{(n)} \sim \pi$, and form the empirical state distribution $\widehat{\rho}$. Consider the greedy skill selector $\widehat{z} := \arg\min_z d(\widehat{\rho}, \rho_z)$. Then*

$$\Pr[\widehat{z} \neq z_\star] \leq 2^S H \exp\left(-\frac{n\Delta^2}{2}\right). \tag{5}$$

*In terms of confidence level $\eta \in (0, 1)$, if*

$$n \geq \frac{2}{\Delta^2}(S \log 2 + \log H - \log \eta), \tag{6}$$

*we have $\Pr[\widehat{z} \neq z_\star] \leq \eta$.*

*Proof.*
**Step 1. A sufficient condition for correct selection.**
Define $\widehat{d} := d(\widehat{\rho}, \rho)$. Triangular inequality gives $d(\widehat{\rho}, \rho_{z_\star}) \leq d(\rho, \rho_{z_\star}) + d(\widehat{\rho}, \rho) = \varepsilon + \widehat{d}$. If

$$\widehat{d} < \frac{\Delta}{2} = \frac{\delta}{2} - \varepsilon, \ \text{ i.e. } \ \delta > 2(\varepsilon + \widehat{d}),$$

then $\widehat{z} = z_\star$ because for every $z \neq z_\star$, by triangular inequality,

$$d(\widehat{\rho}, \rho_z) \geq d(\rho_{z_\star}, \rho_z) - d(\widehat{\rho}, \rho_{z_\star}) \geq \delta - (\varepsilon + \widehat{d}) > \varepsilon + \widehat{d} \geq d(\widehat{\rho}, \rho_{z_\star}).$$

Hence

$$\Pr[\widehat{z} \neq z_\star] \leq \Pr\left[\widehat{d} \geq \frac{\Delta}{2}\right]. \tag{7}$$

**Step 2. Convergence of $\widehat{\rho}$ to $\rho$.**
By Bretagnolle-Huber-Carol (BHC) inequality, for a discrete random variable $X = (X_1, X_2, \ldots, X_k)$,

$$\Pr\left[\sum_{i=1}^k |X_i - np_i| \geq 2\lambda\sqrt{n}\right] \leq 2^k e^{-2\lambda^2}, \quad X \sim \text{Mult}(n, \boldsymbol{p}).$$

Let the state space $\mathcal{S} = \{s_1, \ldots s_S\}$. If we denote $S^{(k)} = (S_1^k, \ldots S_H^k) \in \mathcal{S}^H$, by definition,

$$\widehat{\rho} = \frac{1}{nH} \sum_{k=1}^n \sum_{j=1}^H \mathbf{1}_{S_j^k}.$$

Define $S$-dimension random vectors $X_j = (X_j^1, \ldots X_j^S)$ where $j = 1, \ldots, H$ as $X_j^m = \#\{i : S_j^i = s_m\}$. Then $X_j$ is a $n$-multinomial distribution with

$$\boldsymbol{p}_j = (p_j^1, \ldots p_j^S) = (\rho(S_j = s_1), \ldots, \rho(S_j = s_S)).$$

Thus

$$\widehat{d} = \frac{1}{2}\|\widehat{\rho} - \rho\|_1 = \frac{1}{2}\sum_{s\in\mathcal{S}}|\widehat{\rho}(s) - \rho(s)|$$

$$= \frac{1}{2}\sum_{s\in\mathcal{S}}\left|\frac{1}{nH}\sum_{k=1}^{n}\sum_{j=1}^{H}\mathbf{1}_{S_j^k}(s) - \frac{1}{H}\sum_{j=1}^{H}\rho(S_j = s)\right|$$

$$\leq \frac{1}{2nH}\sum_{j=1}^{H}\sum_{s\in\mathcal{S}}\left|\#\{i : S_j^k = s\} - n\rho(S_j = s)\right|$$

$$= \frac{1}{2nH}\sum_{j=1}^{H}\sum_{m=1}^{S}\left|X_j^m - np_j^m\right|.$$

$$\Pr\left[\widehat{d} \geq \xi\right] \leq \Pr\left[\sum_{j=1}^{H}\sum_{m=1}^{S}\left|X_j^m - np_j^m\right| \geq 2nH\xi\right]$$

$$\leq \Pr\left[\exists j \text{ s.t. } \sum_{m=1}^{S}\left|X_j^m - np_j\right| \geq 2n\xi\right]$$

$$\leq \sum_{j=1}^{H}\Pr\left[\sum_{m=1}^{S}\left|X_j^m - np_j\right| \geq 2n\xi\right]$$

$$\leq 2^S H e^{-2n\xi^2}.$$

Now taking $\xi = \frac{\Delta}{2}$ and combining with equation 7 gives

$$\Pr[\widehat{z} \neq z_\star] \leq 2^S H \exp\left(-\frac{n\Delta^2}{2}\right).$$

In terms of confidence level $\eta \in (0, 1)$,

$$n \geq \frac{2}{\Delta^2}(S\log 2 + \log H - \log \eta) \quad \Rightarrow \quad \Pr[\widehat{z} \neq z_\star] \leq \eta.$$

$\square$

# B ANALYSIS ON ANINFONCE

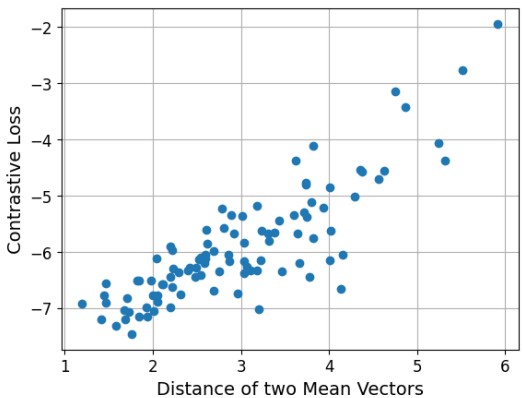

Figure 11: **Effect of mean vector's distance on Loss.** AnInfoNCE objective has a positive correlation with distance between two distributions. Loss was calculated by sampling 1000 points from two 5-dimensional Gaussian distributions.

An analysis was conducted to evaluate the behavior of the AnInfoNCE objective as a function of inter-distribution separation. Two mean vectors were first drawn independently from the standard normal distribution and used to parameterize two Gaussian distributions with identity covariance. AnInfoNCE was then estimated by Monte Carlo sampling from each distribution. Empirically, we observe that the AnInfoNCE loss increases monotonically with the Euclidean distance between the two mean vectors, indicating that larger separations yield higher objective values (Figure 11). This monotonic relationship highlights the ability of AnInfoNCE to promote diversity between learned skills, in contrast to the CeSD objective, which collapses to zero whenever the support sets of the two distributions do not overlap.

## C ADDITIONAL EXPERIMENTS IN MAZE

### C.1 ANALYSIS OF THE EFFECT OF SKILL COUNT

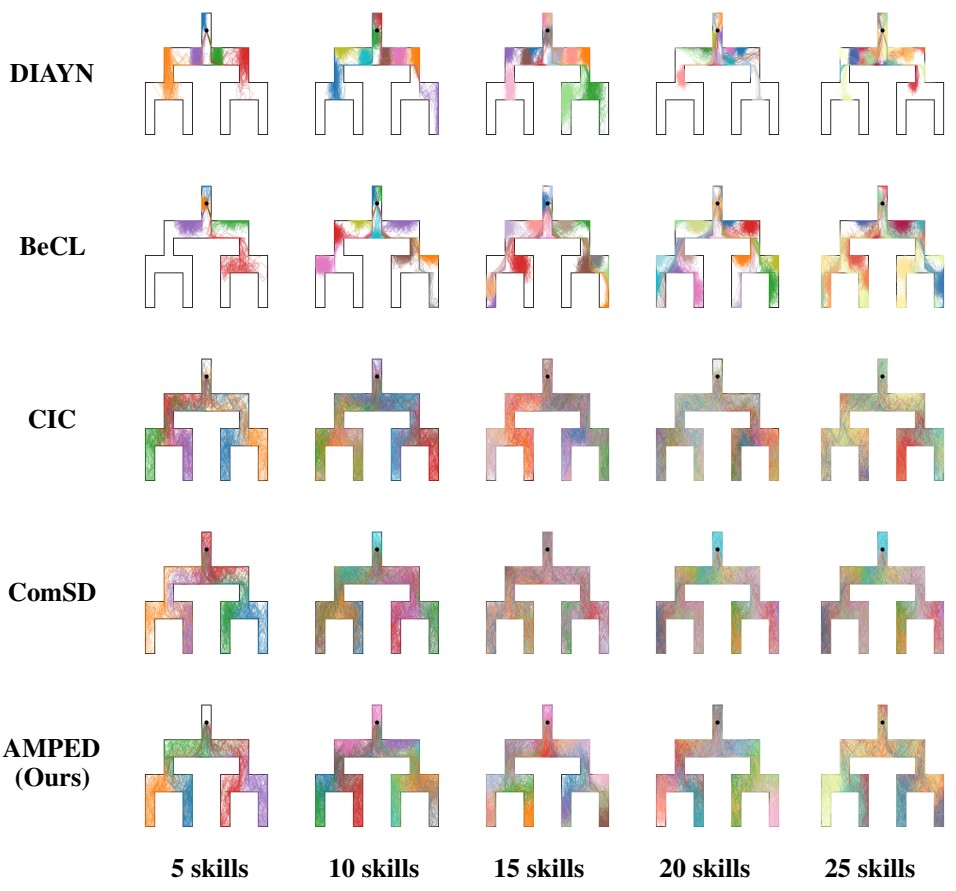

Figure 12: **Skills learned in the Tree Maze under varying skill counts.** This visualization shows the skill allocations of each method as the number of skills increases from 5 to 25. AMPED consistently fills the maze with well-separated regions, whereas DIAYN and BeCL leave gaps, and CIC and ComSD exhibit increasing overlap as the skill count grows.

Figure 12 illustrates how different methods partition the Tree Maze as the number of skills increases from 5 to 25. Unlike DIAYN and BeCL, which tend to leave large regions unexplored or produce overlapping trajectories, our method fills the entire maze while maintaining clear separation between skills. When using 10 or 15 skills, both CIC and ComSD exhibit substantial mixing between skill regions, whereas AMPED preserves distinct, non-overlapping coverage for each skill. At higher skill counts (20 and 25), all methods begin to overlap simply due to capacity limits, making AMPED's advantage over ComSD less visually pronounced. Nonetheless, it still outperforms CIC in maintaining

cleaner skill boundaries. These results confirm that AMPED effectively balances exploration and diversity even as the dimensionality of the skill space grows.

## C.2    COMPARISON ON THE SQUARE MAZE

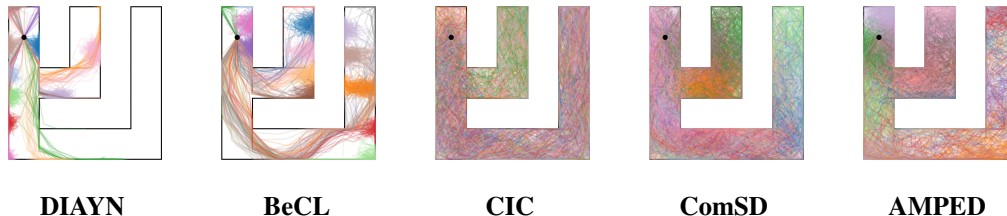

**DIAYN**          **BeCL**          **CIC**          **ComSD**          **AMPED**

Figure 13: **Skills learned in the Square Maze.** Each method's trajectories are shown: AMPED explores every corridor with separated regions, whereas DIAYN and BeCL leave gaps and CIC and ComSD exhibit overlapping trajectories.

In the Square Maze, AMPED achieves full coverage with largely distinct skill regions and only minor overlap (Figure 13). All visualizations use 15 skills per method. By contrast, DIAYN and BeCL leave large areas under-explored or learn only a few broad behaviors, sacrificing either coverage or separation. CIC covers most of the state-space but generates highly entangled trajectories, indicating poor skill disentanglement. ComSD attains coverage similar to AMPED but exhibits more pronounced region mixing. Taken together, these results show that AMPED not only generalize beyond the Tree Maze but both maximizes exploration and enforces strong skill diversity in the Square Maze as well.

## C.3    COMPARISON OF MI AND ENTROPY

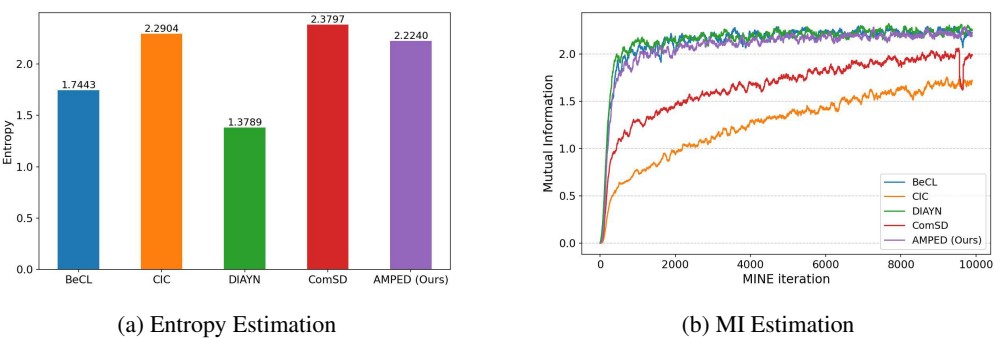

(a) Entropy Estimation                    (b) MI Estimation

Figure 14: **Entropy and MI estimates in the Square Maze.** (a) Particle-based entropy estimates (Liu & Abbeel, 2021b) show that AMPED achieves significantly higher state entropy (exploration) than diversity-focused methods (BeCL, DIAYN), while matching CIC and ComSD. (b) MI estimated via MINE (Belghazi et al., 2018) indicates that AMPED attains diversity comparable to BeCL and DIAYN and substantially exceeds CIC and ComSD.

In the Square Maze, AMPED maximizes both mutual information (MI) and state entropy (Figure 14). To assess whether multi-objective optimization degrades any single objective relative to single-objective training, we compare AMPED's MI and entropy losses against mono-objective baselines. Following BeCL, we use a particle-based entropy estimator and a MINE-based MI estimator, evaluating 10 skills in the Square Maze (Belghazi et al., 2018). Taken together, these results indicate that AMPED simultaneously maximizes exploration and diversity.

## C.4    EVOLUTION OF SKILLS ACROSS TIME STEPS

As illustrated in Figure 15, early in training (epoch 1), the policy aggressively explores new branches, rapidly expanding its state coverage. By epoch 3 and 5, the trajectory has spanned nearly the entire

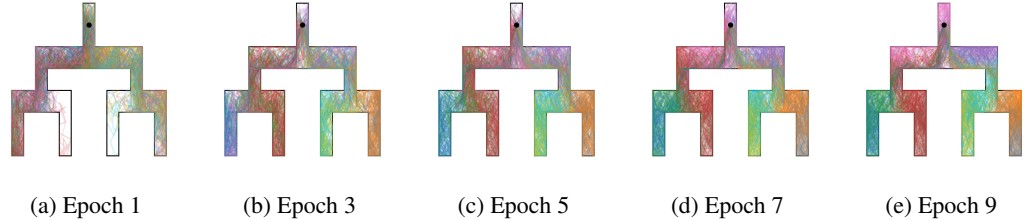

| (a) Epoch 1 | (b) Epoch 3 | (c) Epoch 5 | (d) Epoch 7 | (e) Epoch 9 |

Figure 15: **Skill trajectory evolution in the Tree Maze over training epochs.** A representative skill's path is shown at 2-epoch intervals, illustrating initial rapid expansion of coverage followed by progressive refinement into distinct, well-separated skill regions.

maze, maximizing exploration. In later stages (epoch 7 and 9), the skill's path is refined: it begins to adjust which corridors it traverses to carve out distinct regions and increase diversity. Although such clear visualizations are not possible in high-dimensional spaces, this simple sequence provides intuition for how our method first drives broad exploration and then sculpts well-separated skill behaviors.

# D    ABLATION STUDY ON REWARD SCALING FACTOR

Table 2: **Performance comparison under extreme $\alpha$ and $\beta$ settings.** $\alpha$ and $\beta$ control the relative weight of entropy-based and RND rewards. AMPED (Ours) result are computed as return (mean $\pm$ standard deviation) over 10 random seeds, while each $\alpha$ or $\beta$ configuration is evaluated using three random seeds. The best result is shown in **bold**, and the second-best is underlined.

| Domain | Task | AMPED (Ours) | $\alpha = 0$ | $\alpha = 100$ | $\beta = 0$ | $\beta = 1000$ |
|---|---|---|---|---|---|---|
| Walker | Flip | **674** $\pm$ 105 | 609 $\pm$ 44 | 587 $\pm$ 74 | 524 $\pm$ 38 | 586 $\pm$ 88 |
| | Run | 467 $\pm$ 103 | **505** $\pm$ 10 | 420 $\pm$ 99 | 382 $\pm$ 29 | **505** $\pm$ 39 |
| | Stand | 951 $\pm$ 38 | 942 $\pm$ 26 | **956** $\pm$ 6 | 948 $\pm$ 9 | 923 $\pm$ 30 |
| | Walk | **929** $\pm$ 19 | 913 $\pm$ 43 | 908 $\pm$ 21 | 863 $\pm$ 59 | 878 $\pm$ 94 |
| | Sum | **3021** | 2969 | 2871 | 2717 | 2892 |
| Quadruped | Jump | **720** $\pm$ 34 | 677 $\pm$ 50 | 710 $\pm$ 59 | 623 $\pm$ 100 | 648 $\pm$ 61 |
| | Run | 494 $\pm$ 56 | 493 $\pm$ 23 | 459 $\pm$ 122 | 371 $\pm$ 127 | **613** $\pm$ 92 |
| | Stand | **906** $\pm$ 71 | 865 $\pm$ 55 | 837 $\pm$ 117 | 904 $\pm$ 27 | 875 $\pm$ 23 |
| | Walk | **890** $\pm$ 62 | 844 $\pm$ 58 | 805 $\pm$ 73 | 720 $\pm$ 118 | 852 $\pm$ 53 |
| | Sum | **3010** | 2879 | 2811 | 2618 | 2998 |
| Jaco | Re. bottom left | **143** $\pm$ 34 | 94 $\pm$ 31 | 123 $\pm$ 22 | 133 $\pm$ 35 | 134 $\pm$ 16 |
| | Re. bottom right | **144** $\pm$ 27 | 111 $\pm$ 15 | 96 $\pm$ 15 | 114 $\pm$ 35 | 118 $\pm$ 37 |
| | Re. top left | 140 $\pm$ 41 | 115 $\pm$ 18 | **159** $\pm$ 24 | 145 $\pm$ 63 | 121 $\pm$ 3 |
| | Re. top right | **154** $\pm$ 49 | 112 $\pm$ 22 | 143 $\pm$ 18 | 112 $\pm$ 16 | 106 $\pm$ 11 |
| | Sum | **581** | 432 | 521 | 504 | 479 |

In this ablation, we keep every setting in Appendix I.5 fixed except for one of the reward-scaling factors. As shown in Table 2, deviating from the defaults on either $\alpha$ or $\beta$ degrades the sum of episode returns in most domains. Setting $\alpha = 0$ (no entropy reward) or $\beta = 0$ (no RND reward) leads to substantial drops, while excessively large values ($\alpha = 100$ or $\beta = 1000$) improve some individual tasks but hurt overall performance. The default AMPED weights achieve the best aggregate performance, underscoring the need for balanced scaling between exploration and novelty signals.

In Table 3, we present hyperparameter search results for the entropy reward weight $\alpha$ and the RND reward weight $\beta$. In the Walker domain, the average sum return across five $(\alpha, \beta)$ configurations is 2989.4, which still ranks second among the baselines in Table 13. Notably, the $(0.01, 8)$ setting achieves a sum return of 3323, outperforming all baselines. Similarly, in the Quadruped domain, the

Table 3: **Hyperparameter search results for $\alpha$ and $\beta$.** Results reflect single-run returns for each modified configuration, except for AMPED. The best result is shown in **bold**, and the second-best is underlined.

| Domain | Task | **AMPED** (0.01, 10) | (0.02, 10) | (0.005, 10) | (0.01, 12) | (0.01, 8) |
|---|---|---|---|---|---|---|
| Walker | Flip | $\underline{674} \pm 105$ | 667 | 535 | 599 | **886** |
| | Run | $467 \pm 103$ | 451 | 354 | $\underline{550}$ | **565** |
| | Stand | $951 \pm 38$ | $\underline{960}$ | 942 | **964** | 949 |
| | Walk | $\mathbf{929} \pm 19$ | 893 | 891 | 797 | $\underline{923}$ |
| | Sum | $\underline{3021}$ | 2971 | 2722 | 2910 | **3323** |
| | Task | **AMPED** (0.002, 8) | (0.004, 8) | (0.001, 8) | (0.002, 10) | (0.002, 6) |
| Quadruped | Jump | $\mathbf{720} \pm 34$ | 597 | 702 | 710 | $\underline{717}$ |
| | Run | $494 \pm 56$ | $\underline{499}$ | 276 | **512** | 488 |
| | Stand | $906 \pm 71$ | **956** | 792 | $\underline{918}$ | $\underline{918}$ |
| | Walk | $\underline{890} \pm 62$ | 833 | 853 | **891** | 869 |
| | Sum | $\underline{3010}$ | 2885 | 2623 | **3031** | 2992 |

average across the five configurations is 2908.2, exceeding the best baseline performance. Moreover, the $(0.002, 10)$ configuration yields even better performance than our default hyperparameter choice. These results suggest that careful tuning of $\alpha$ and $\beta$ can yield further improvements for AMPED.

## E  ANALYSIS ON SKILL SELECTION

One of the central motivations for our approach is that the skill selector, responsible for choosing among a diverse, pretrained set of skills, should substantially enhance overall performance. However, an ablation in the Walker domain (3021 with the selector vs. 3036 without it) revealed that adopting the skill selector does not lead to consistent improvements.

To investigate this discrepancy, we designed a complementary experiment isolating each pretrained skill. For each task, we fix a single skill and condition the policy exclusively on that skill during fine-tuning, both for training and evaluation. We conducted three-seed evaluations on Walker-flip, run, where the selector had previously degraded performance, and on Quadruped-stand, walk-where it had been beneficial (3,010 with the selector vs. 2,824 without).

Table 4: **Fine-tuning returns under different skill-selection regimes.** "Single-Skill" reports the average return across fine-tuning each pretrained skill individually; "Oracle Best Skill" denotes the highest return achieved by the single best skill. Results are computed over three random seeds and reported as mean $\pm$ standard deviation. The best result is shown in **bold**, and the second-best is underlined.

| Domain | Task | Skill Selector | Random Skill | Single-Skill | Oracle Best Skill |
|---|---|---|---|---|---|
| Walker | Flip | $674 \pm 105$ | $686 \pm 133$ | $\underline{719} \pm 121$ | $\mathbf{913} \pm 3$ |
| | Run | $467 \pm 103$ | $\underline{517} \pm 49$ | $503 \pm 74$ | $\mathbf{603} \pm 19$ |
| Quadruped | Stand | $906 \pm 71$ | $816 \pm 150$ | $\underline{911} \pm 43$ | $\mathbf{959} \pm 5$ |
| | Walk | $\underline{890} \pm 62$ | $816 \pm 116$ | $837 \pm 62$ | $\mathbf{912} \pm 17$ |

The fixed-skill results in Table 4 show that the skill selector underperforms single-skill fine-tuning across three tasks. Crucially, this deficit persists whether the selector had previously appeared beneficial (as in the Quadruped environments) or not (as in Walker), indicating that the selector itself, rather than domain-specific factors, is the primary bottleneck. Although dedicated fine-tuning benefits from sustained gradient updates targeted at a single skill, the observed performance gap suggests that the selector's learning is hampered, likely by sparse rewards. Consequently, its value estimates

Table 5: **Number of unique skills used per task.** Results are computed over three random seeds and reported as mean ± standard deviation.

| Domain | Task | Unique Skills Used |
|---|---|---|
| Walker | Flip | $4.00 \pm 1.73$ |
| | Run | $3.67 \pm 1.53$ |
| Quadruped | Stand | $2.67 \pm 0.58$ |
| | Walk | $4.33 \pm 0.58$ |

remain unstable, leading to suboptimal choices. These findings underscore the need for more robust training strategies for the skill selector.

To assess the stability and consistency of the skill selector, we measure the number of distinct skills invoked during the last 5K fine-tuning episodes for each downstream task. As summarized in Table 5, the selector consistently concentrates on a small subset of available skills, particularly toward the end of training. This pattern suggests convergence to a stable, task-specific mapping from observations to high-level skills, underscoring the utility and reusability of the learned skill space.

## F    RELATED WORKS

### F.1    UNSUPERVISED REINFORCEMENT LEARNING

URL aims to train general-purpose policies capable of rapid adaptation to diverse downstream tasks. This is achieved through the design of intrinsic objectives or rewards that guide exploration without relying on explicit external feedback. URL typically involves two stages: (1) pretraining, where agents develop foundational behaviors driven by intrinsic motivation, and (2) fine-tuning, where these behaviors are adapted to task-specific objectives.

URLB (Laskin et al., 2021) categorizes existing URL algorithms into three primary groups:

1. Data-based approaches encourage agents to explore novel states by maximizing state entropy, fostering diverse experiences during pretraining. Notable methods include APT (Liu & Abbeel, 2021b), which utilizes particle-based entropy estimators to maximize the distance between k-nearest neighbors (kNN) in observation embeddings. ProtoRL (Yarats et al., 2021) builds on this idea by incorporating prototypical representation learning, inspired by SWaV (Caron et al., 2020), to enhance exploration efficiency. CIC (Laskin et al., 2022) extends ProtoRL by introducing skills, positioning CIC as both a data-based and competence-based method.

2. Knowledge-based approaches aim to improve an agent's understanding of environmental dynamics by maximizing prediction errors, thus incentivizing the exploration of novel or poorly understood states. The Intrinsic Curiosity Module (ICM) (Pathak et al., 2017) encourages exploration by rewarding agents based on the error in predicting future state transitions. Reyes et al. (2022) extended this idea by incorporating the prediction of joint observations. On the other hand, disagreement-based methods (Pathak et al., 2019) quantify uncertainty through an ensemble of predictive models, rewarding states where model predictions diverge significantly. Random Network Distillation (RND) (Burda et al., 2019) measures novelty via the prediction error of a random, fixed target network, where higher errors indicate unfamiliar states. Nikulin et al. (2023) enhanced this idea by applying Feature-wise Linear Modulation.

3. Competence-based approaches, often referred to as unsupervised skill discovery, seek to develop a diverse repertoire of skills without relying on external rewards. These methods are grounded in information-theoretic principles, typically maximizing MI between skill embeddings and state, or trajectories to ensure meaningful and diverse behaviors. For instance, VIC (Gregor et al., 2016) maximizes controllability of skills by setting MI between skills and final state, given the initial state as an objective. DIAYN (Eysenbach et al., 2019) encourages diversity by maximizing MI between skills and states while ensuring skills are distinguishable. BeCL (Yang et al., 2023) leverages contrastive learning to enhance skill discriminability by maximizing MI between trajectories generated from the same skill; this also has a side effect that maximizes the entropy in the limiting case.

Our approach synthesizes principles from data-based, knowledge-based, and competence-based methods, drawing on models such as CIC (Laskin et al., 2022), RND (Burda et al., 2019), CeSD (Bai et al., 2024), and BeCL (Yang et al., 2023). Specifically, we address the limitations of these models in balancing exploration and skill diversity by introducing novel methods for integrating them.

### F.2 Unsupervised Skill Discovery

Competence-based approaches, commonly referred to as unsupervised skill discovery, have garnered significant attention in recent years due to their potential to enable agents to acquire diverse, discriminative behaviors without external supervision. It focuses on enabling agents to learn distinct, discriminating behaviors without external supervision. Skill diversity has been shown to be critical for downstream task performance, both empirically and theoretically (Eysenbach et al., 2019; Laskin et al., 2022; Yang et al., 2024). This is often achieved by maximizing the MI between states or trajectories with skills, encouraging agents to develop diverse and meaningful behaviors. Key contributions in this area include works by (Gregor et al., 2016; Florensa et al., 2017; Eysenbach et al., 2019; Sharma et al., 2020; Baumli et al., 2021).

However, these studies (Campos et al., 2020; Strouse et al., 2022; Park et al., 2022) have highlighted limitations in traditional MI-based methods, noting that maximizing MI between states and skills can lead to suboptimal exploration. It is also theoretically shown that such approach can not construct an optimal policy for some downstream tasks (Eysenbach et al., 2022; Yang et al., 2024). There are some methods to address this when the observation space is Cartesian coordinate space (Park et al., 2022; Zhao et al., 2021); while effective in specific navigation tasks, these approaches impose strong assumptions and are less adaptable to general situation. To address these limitations, alternative approaches introduce auxiliary exploration mechanisms and refined training techniques aimed at enhancing exploration. While many methods focus on modifying the objective functions, such as DIAYN, BeCL, CeSD, ComSD, and CSD, others explore architectural innovations and dynamic exploration strategies, as seen in DSG (Bagaria et al., 2021), EDL (Campos et al., 2020), and ReST (Jiang et al., 2022). These techniques aim to promote diverse exploration without relying solely on objective modifications.

## G Difference with Former Studies

Prior to our work, two representative methods for jointly considering exploration and diversity are CeSD and ComSD. However, our method departs from these approaches in several important ways, which will be explained in detail. Note that on URLB, our approach achieves 20.91% and 35.01% higher returns than CeSD and ComSD, respectively.

### G.1 Difference with CeSD

Instead of diversifying skills using MI, CeSD maximizes exploration using the entropy, while adding a regularization term for diversifying skills. This approach mitigates the paucity of exploration while simultaneously accounting for a diverse array of skills. Unfortunately, the algorithm is time-consuming because it includes clustering states. The paper avoids this bottleneck by choosing a subset of states for clustering, which would lead to inaccurate estimation of clustering and, therefore, instability of training. Also, their diversity regularization becomes ineffective once the state distributions of different skills no longer intersect. This can be seen from their auxiliary reward:

$$r_i^{\text{reg}} = \frac{1}{|\mathbb{S}_i^{\text{pe}} - \mathbb{S}_i^{\text{clu}}| + \lambda}.$$

Here, $\mathbb{S}_i$ denotes the state coverage set of skill $i$, $\mathbb{S}_i^{\text{pe}}$ is the expanded coverage obtained during the exploratory phase, and $\mathbb{S}_i^{\text{clu}}$ is the clustered coverage set after applying clustering to the *union* of all collected $\mathbb{S}_i^{\text{pe}}$ across skills. Because clustering is applied to this union, once the supports of different skills are already disjoint, the diversity term effectively saturates: the clustering no longer distinguishes between "slightly separated" and "very far apart" supports. In this regime, the regularizer ceases to provide a meaningful gradient that encourages further separation or broader coverage, and skills can end up concentrating on a limited portion of the state space even when other regions remain under-explored.

Such an effect can be seen in the 2D maze experiment; other methods like BeCL or DIAYN separate skills effectively, while CeSD does not. This may be advantageous in low dimension environment like a 2D maze because one can fully cover the whole space. However, in high-dimensional domains such as URLB, insufficient separation of skills degrades downstream task performance, as established by our Theorem 1.

### G.2 DIFFERENCE WITH COMSD

Similar to ComSD, our approach aims to balance the diversity and exploration objectives. ComSD uses the entropy of trajectory $H(\tau)$ as a exploration objective, and negative entropy of trajectory conditioned to skill $-H(\tau|z)$ as a diversity objective. $H(\tau)$ is estimated using a particle-based approach and $H(\tau|z)$ is estimated using a variational approach. To balance exploration and diversity, ComSD employs a specialized weighting mechanism called Skill-based Multi-objective Weighting (SMW), which assigns different optimization objectives to different skills; some skills emphasizing diversity while others prioritize exploration. However, this selective assignment does not necessarily lead to optimal overall performance. ComSD's method merely differentiates each skill's repulsiveness from others, which does not guarantee an ideal trade-off between exploration and diversity. Moreover, it lacks a solid theoretical foundation to justify the weighting strategy.

In contrast, our method explicitly aims to maximize both exploration and diversity, grounded in the concept of gradient conflict, which has been extensively studied in prior research (Yu et al., 2020; Liu et al., 2021; Navon et al., 2022). By directly addressing the conflicts between exploration and diversity gradients, our method achieves a more stable and theoretically justified optimization process.

## H OBJECTIVES AND REWARDS

Table 6: **Comparison of algorithms.** Intrinsic Objectives and Rewards of each methods are shown.

| Algorithm | Intrinsic Objective | Intrinsic Reward $(r_t^i)$ |
|---|---|---|
| APT | $H(\phi(s))$ | $r^{part}$ |
| ICM | $\mathbb{E}_\tau[\sum_t \gamma^t r_t^i]$ | $\frac{\eta}{2}\|\hat{\phi}(s_{t+1}) - \phi(s_{t+1})\|_2^2$ |
| RND | $\mathbb{E}_\tau[\sum_t \gamma^t r_t^i]$ | $\|\hat{f}(x_t) - f(x_t)\|^2$ |
| CIC | $H(\tau) - H(\tau|z)$ | $r^{part} + \log q(\tau|z)$ |
| DIAYN | $H(z) - H(z|s) + H(a|s,z)$ | $\log q(z|s) - \log p(z)$ |
| DADS | $H(s'|s) - H(s'|s,z)$ | $\log q(s'|s,z) - \log \sum_{i=1}^K q(s'|s,z_i)$ |
| BeCL | $I(s^{(1)}; s^{(2)})$ | $r^{contr}$ |
| CeSD | $H(s) + \alpha \cdot \sum_{s \in \mathcal{S}} |d^{\pi_i}(s) - d^{\hat{\pi}_i}(s)|$ | $r^{part} + \alpha/(|S_i^{pe} \backslash S_i^{clu}| + \lambda)$ |
| ComSD | $H(\tau) - H(\tau|z)$ | $r^{part} + \alpha \cdot r^{contr}$ |
| **AMPED (Ours)** | $\alpha \cdot H(s) + \beta \cdot \mathcal{L}_{RND} + I(s^{(1)}; s^{(2)})$ | $\alpha \cdot r^{part} + \beta \cdot \|\hat{f}(x_t) - f(x_t)\|^2 + r^{AnInfo}$ |

$z$ denote a skill, $\phi(s)$ denote a encoded state, and $\text{NN}_k$ denote a $k$ nearest neighbor. Neglected the loss for training state encoders. $r^{part}$ is a particle-based entropy estimation, and $r^{contr}$ is a contrastive-based MI estimation; the specific reward varies slightly depending on the method. The canonical MI objective by InfoNCE is defined as:

$$r^{part} = \sum_{i=1}^n \log \|z_i - \text{NN}_k(z_i)\|, \quad r^{contr} = \mathbb{E}\left[\frac{\exp(f(s_t^{(1)}) \cdot f(s_t^{(2)})/\kappa)}{\sum_{s_j \in S^{-1} \cup \{s_t^{(2)}\}} \exp(f(s_t^{(1)}) \cdot f(s_j)/\kappa)}\right] \quad (8)$$

## I IMPLEMENTATION DETAILS

### I.1 MAZE ENVIRONMENTS

The maze environments are adapted from the open-source EDL implementation by (Campos, 2020). In this setup, the observation is given as $\mathcal{S} \in \mathbb{R}^2$, which represents the current position, while the

Table 7: **Environment detail of square-maze used for evaluation.**

| Parameter | Value |
|---|---|
| State space | $\mathcal{S} \in \mathbb{R}^2$ |
| Action space | $\mathcal{A} \in [-0.95, 0.95]^2$ |
| Episode length | 50 |
| Size: Square maze (Figure 13) | $5 \times 5$ |
| Size: Tree maze (Figure 7 and Figure 12) | $7 \times 7$ |

action is given as $\mathcal{A} \in \mathbb{R}^2$, corresponding to the velocity and direction. The agent observes only its current position and does not have access to the locations of walls, which must be inferred through interaction with the environment. At the start of each episode, the agent's initial state is uniformly sampled within a $1 \times 1$ tile. Table 7 summarizes the details and topological characteristics of each maze used in the experiments.

## I.2 NETWORK ARCHITECTURES FOR MAZE EXPERIMENTS

All code was based on the open-source EDL implementation by Campos (2020). We used PPO (Schulman et al., 2017) as our on-policy algorithm, with both policy and value functions parameterized by three hidden layers of size 128 and ReLU activations. The policy network takes the concatenated state and goal vectors, passes them through three 128-unit MLP layers, then applies a tanh output scaled by the action range. The critic shares the same three-layer backbone but outputs a single scalar Q-value given $(s, a)$.

For intrinsic rewards, we employed three specialized networks: a CIC encoder comprising a state network that maps the state vector through two 128-unit hidden layers to an $n$-dimensional embedding and a predictor network that takes the concatenated pair of these embeddings (size $2n$), processes it through two 128-unit hidden layers, and outputs an $n$-dimensional prediction; an RND network comprising predictor and frozen target MLPs (each with two 128-unit hidden layers) mapping observations to a n-dimensional feature space, where the mean squared prediction error defines $r_{\text{rnd}}$; and a BeCL encoder, a three-layer 128-unit MLP that maps observations to an $n$-dimensional skill embedding for the AnInfoNCE loss, encouraging non-overlapping skill distributions.

## I.3 URLB ENVIRONMENTS

The Walker domain focuses on training a biped constrained to a 2D vertical plane to acquire balancing and locomotion skills (Laskin et al., 2021). It includes four downstream tasks: *Stand*, *Walk*, *Flip*, and *Run*. The observation space is defined as $\mathcal{S} \in \mathbb{R}^{24}$, and the action space as $\mathcal{A} \in \mathbb{R}^6$.

The Quadruped domain involves training a four-legged agent for balance and locomotion within a 3D environment. This domain includes four tasks (Figure 16): *Stand*, *Walk*, *run*, and *flip*. The observation space is $\mathcal{S} \in \mathbb{R}^{78}$, and the action space is $\mathcal{A} \in \mathbb{R}^{16}$.

The Jaco domain is designed for manipulation tasks using a 6-DoF robotic arm equipped with a three-finger gripper. It includes four tasks: *Reach Top Left*, *Reach Top Right*, *Reach Bottom Left*, and *Reach Bottom Right*. The observation space is $\mathcal{S} \in \mathbb{R}^{55}$, and the action space is $\mathcal{A} \in \mathbb{R}^9$.

## I.4 NETWORK ARCHITECTURES FOR URLB EXPERIMENTS

This section describes the network architecture of our method. At the beginning of each episode, a skill vector $z$ is sampled, where the default setting uses a one-hot encoding with skill_dim $= 16$. This skill vector is concatenated with the processed observation features and used as input to the policy, value, and intrinsic reward modules.

The raw observation is first processed by a four-layer convolutional encoder, where each layer has 32 channels and uses Rectified Linear Unit (ReLU) activations. The encoder's output feature map is then flattened into a latent feature vector of dimensions (repr_dim $= 32 \times 35 \times 35$). The resulting feature vector is combined with the skill vector before being forwarded to the downstream networks.

**Walker**          **Quadruped**          **Jaco**

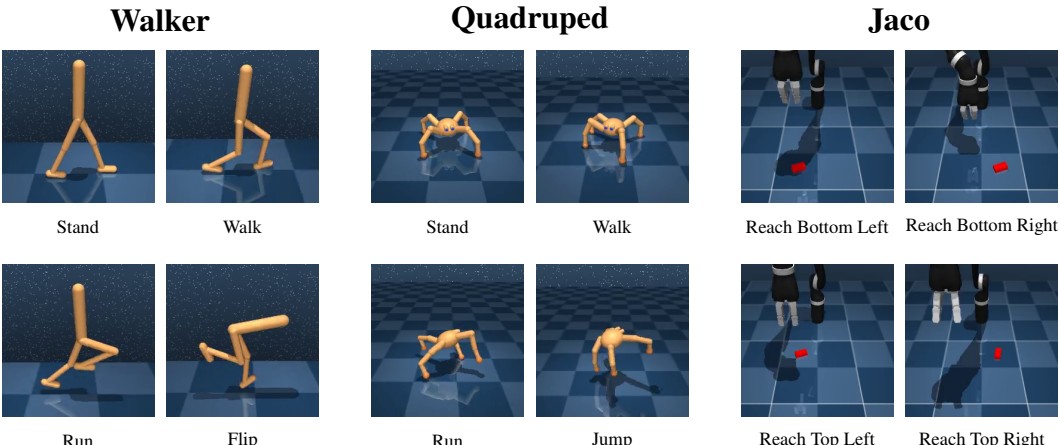

Figure 16: **Visualization of the downstream tasks: Walker, Quadruped, and Jaco domains.**

For decision making, the actor network processes this combined observation and skill vector through a trunk consisting of a linear layer, layer normalization, and hyperbolic tangent (Tanh) activations. The resulting feature is passed through two fully connected layers with 1024 hidden dimensions and ReLU activations, and finally projected to the action space to produce the action distribution. The critic network applies the same trunk structure to the combined representation. It then concatenates the resulting features with the action, and processes them through two additional hidden layers with 1024 dimensions each to estimate the Q-values.

The RND module constructs a predictor-target architecture by copying the observation encoder. Both the predictor and the frozen target network share the same initial encoder and are extended with a multilayer perceptron (MLP) composed of two hidden layers of 1024 dimensions. The predictor is trained to minimize the mean-squared error relative to the target's output.

The CIC module integrates three coordinated components, which consist of a state encoder that transforms both the current and next observation features into embeddings within the skill space, a skill projection network that embeds the sampled skill vector, and a predictor network that takes the concatenated state and next-state embeddings and transforms them into a representation aligned with the skill embedding. All components use MLP with hidden layers of 1024 dimensions, and the module is optimized using a contrastive predictive coding (CPC) objective to encourage alignment between state transitions and the correct skill representation. The state encoder's outputs are used to calculate the $r_{entropy}$, based on kNN distances.

The BeCL module takes observation features (excluding the skill vector) and processes them through an embedding network with two hidden layers of 1024 dimensions to produce a compact representation. This representation is then passed through a projection head, which includes another hidden layer with 1024 dimensions and an output layer, producing outputs that match the skill dimension. This layered architecture enables the module to generate embeddings that are optimized for contrastive learning, effectively encouraging skill discrimination in the learned space.

During fine-tuning, we employ a Soft Actor-Critic (SAC)-based skill selector to adaptively choose a skill vector given the current observation. SAC offers off-policy sample efficiency and entropy-regularized stability, which help balance exploration and exploitation. The skill selector consists of a policy network and a value network. The policy network maps observations to a discrete distribution over skills. It consists of a linear layer follwed by layer normalization and a Tanh activation, then two fully connected layers with 256 hidden dimensions and ReLU activations, producing logits over the skill space. A skill is sampled from this distribution using an epsilon-greedy strategy. The critic network use the same input processing as the policy network. It then maps the resulting feature through two fully connected layers with 256 hidden dimensions to produce Q-values for each skill. During training, the critic is updated using temporal difference learning, while the actor is optimized via entropy-regularized policy gradients (Haarnoja et al., 2018).

In the $\epsilon$-greedy skill selection strategy, the exploration probability $\epsilon$ decays exponentially over time, starting from $\epsilon = 1.0$ and gradually decreasing to $\epsilon = 0.01$ with an decay factor of 20000

steps. This encourages early-stage exploration of diverse skills and gradually shifts toward selecting high-performing skills.

## I.5  HYPERPARAMETERS

Table 8 and Table 9 contains the hyperparameters we use. Hyperparameter values for the Maze environment were adopted directly from the EDL repository (Campos, 2020), while those for the URLB environment follow the defaults provided by the URLB codebase (Laskin & Yarats, 2025). We perform hyperparameter tuning in URLB, focusing on three key components, intrinsic reward weights $(\alpha, \beta)$, the projection probability $p$. We explored values in the ranges $\alpha \in [10^{-3}, 0.1]$, $\beta \in [10^{-3}, 10]$, $p \in [0.5, 1.0]$.

Table 8: **Hyperparameter settings for URLB experiments.**

| Intrinsic reward hyperparameter | Walker | Quadruped | Jaco |
|---|---|---|---|
| skill dimension | 16 | 16 | 16 |
| contrastive update rate | 3 | 3 | 3 |
| temperature | 0.5 | 0.5 | 0.5 |
| alpha ($\alpha$) | 0.01 | 0.002 | 0.03 |
| beta ($\beta$) | 10 | 8 | 0.005 |
| projection probability ($p$) | 0.6 | 0.65 | 0.8 |
| Number of nearest neighbors ($k$) | 16 | 16 | 16 |
| **Skill selector hyperparameter** | **Value** | | |
| epsilon start | 1.0 | | |
| epsilon end | 0.01 | | |
| epsilon step | 20000 | | |
| learning rate | $3 \times 10^{-4}$ | | |
| **DDPG hyperparameter** | **Value** | | |
| replay buffer capacity | $10^6$ | | |
| warmup frames | 4000 | | |
| $n$-step returns | 3 | | |
| mini-batch size | 1024 | | |
| discount ($\gamma$) | 0.99 | | |
| learning rate | $10^{-4}$ | | |
| agent update frequency | 2 | | |
| critic target EMA rate ($\tau_Q$) | 0.01 | | |
| exploration stddev clip | 0.3 | | |
| exploration stddev value | 0.2 | | |
| number of pre-training frames | $2 \times 10^6$ | | |
| number of fine-tuning frames | $1 \times 10^5$ | | |

Table 9: **Hyperparameter settings for Tree-maze experiments.**

| HyperParameter | Value |
|---|---|
| learning rate ($\tau$) | 3e-4 |
| discount ($\gamma$) | 0.99 |
| GAE lambda | 0.98 |
| entropy lambda | 0.025 |
| hidden dim | 128 |
| temperature | 0.5 |
| alpha ($\alpha$) | 0.01 |
| beta ($\beta$) | 1e-4 |
| projection probability | 0.5 |
| knn k | 16 |
| knn clip | 5e-4 |
| epoch | 50 |

## I.6 URLB TRAINING PIPELINE

---

**Algorithm 1** Gradient Surgery

---

1: **Given:** $\nabla\mathcal{L}_{\text{diversity}}$, $\nabla\mathcal{L}_{\text{exploration}}$, probability $p$, parameter of critic $\theta_{\text{critic}}$, and a learning rate $\eta$.
2: **if** $\sum(\nabla\mathcal{L}_{\text{diversity}} \cdot \nabla\mathcal{L}_{\text{exploration}}) < 0$ **then**
3:      With probability $p$:
4:          $\nabla\mathcal{L}_{\text{diversity}} \leftarrow \text{Proj}_{\nabla\mathcal{L}_{\text{exploration}}^{\perp}}(\nabla\mathcal{L}_{\text{diversity}})$
5:      Otherwise:
6:          $\nabla\mathcal{L}_{\text{exploration}} \leftarrow \text{Proj}_{\nabla\mathcal{L}_{\text{diversity}}^{\perp}}(\nabla\mathcal{L}_{\text{exploration}})$
7: **end if**
8: $\Delta\theta_{\text{critic}} = \eta(\nabla\mathcal{L}_{\text{diversity}} + \nabla\mathcal{L}_{\text{exploration}})$
9: $\theta_{\text{critic}} \leftarrow \theta_{\text{critic}} - \Delta\theta_{\text{critic}}$

---

**Algorithm 2** Unsupervised Pretraining with Intrinsic Rewards and Gradient Surgery

---

1: **Given:** number of skills $n$, pretraining frames $N_{\text{PT}}$, seed frames $T$, batch size $N$, update interval $N_{\text{update}}$, policy $\pi_\theta$, critic $Q_\psi$
2: **Initialize:** replay buffer $\mathcal{B} \leftarrow \emptyset$, timestep $t \leftarrow 0$
3: **while** $t < N_{\text{PT}}$ **do**
4:      **if** $t \bmod N_{\text{update}} = 0$ **then**
5:          Sample skill $z_t \sim \text{Uniform}[1, n]$
6:      **end if**
7:      Collect transition $(s_t, a_t, s_{t+1}) \sim \pi_\theta(\cdot \mid s_t, z_t), p(s_{t+1} \mid s_t, a_t)$
8:      store $(s_t, a_t, s_{t+1}, z_t)$ in $\mathcal{B}$
9:      **if** $t \geq T$ **then**                        ▷ begin intrinsic-reward pretraining
10:          Sample batch $\{(s, a, s', z)\}_{i=1}^{N} \sim \mathcal{B}$
11:          **Update encoders:**
12:              Minimize contrastive loss (Eq. 3), RND prediction loss, and AnInfoNCE loss (Eq. 4)
13:          **Compute intrinsic rewards:**
14:              Calculate $r_{\text{exploration}}$, $r_{\text{diversity}}$ as defined in Sec. 3
15:          **Update critic & actor:**
16:              Compute gradients for diversity and exploration losses
17:              Apply Gradient Surgery (Alg. 1)
18:              Update policy $\pi_\theta$ and critic $Q_\psi$
19:      **end if**
20:      $t \leftarrow t + 1$
21: **end while**

---

**Algorithm 3** Fine-Tuning with Extrinsic Rewards and Joint Skill Selector Training

---

1: **Given:** number of finetuning frames $N_{\text{FT}}$, batch size $N$, update interval $U$, critic $Q_\psi$, pretrained policy $\pi_\theta$, skill selector $p_\phi(z \mid s)$
2: **Initialize:** replay buffer $\mathcal{D} \leftarrow \emptyset$, timestep $t \leftarrow 0$
3: **while** $t < N_{\text{FT}}$ **do**
4:      Observe state $s_t$
5:      Select skill $z_t \sim p_\phi(z \mid s_t)$
6:      Select action $a_t \sim \pi_\theta(a \mid s_t, z_t)$
7:      Execute $a_t$ to obtain $(s_{t+1}, r_t)$
8:      Store $(s_t, a_t, r_t, s_{t+1}, z_t)$ in $\mathcal{D}$
9:      **if** $t \bmod U = 0$ **then**                    ▷ Update both selector and agent
10:          Sample batch $\{(s, a, r, s', z)\}_{i=1}^{N} \sim \mathcal{D}$
11:          Update $\theta, \psi$ using extrinsic reward $r$
12:      **end if**
13:      $t \leftarrow t + 1$
14: **end while**

---

### I.7 TRAINING TIME COMPARISON

Table 10 compares the wall-clock training time of AMPED against a range of baselines on the Walker, Quadruped, and Jaco domains. Notably, AMPED incurs only a modest increase in runtime relative to competitive methods like CeSD and BeCL despite exceeding their downstream performance (see Figure 8). Although AMPED requires more computation than CIC (an overhead of 3-6 hours), this extra cost yields substantial performance gains over CIC's purely entropy-based exploration. Overall, these findings demonstrate that AMPED strikes a favorable balance between computational cost and empirical performance.

Table 11 reports fine-tuning times. Because AMPED (Ours) includes the SAC-based skill selector, its fine-tuning incurs a modest overhead of approximately 0.06-0.11 h (4-7 min) compared to baselines such as CIC and BeCL. In future work, we aim to further optimize runtime efficiency, perhaps via more streamlined encoder updates or low-precision training, while preserving the joint handling of exploration and diversity.

Table 10: **Pretraining time (hours with decimal minutes) comparison across baselines.** Results are computed over 10 random seeds and reported as mean $\pm$ standard deviation.

| Domain | AMPED (Ours) | CeSD | CIC | BeCL | APT | RND | DIAYN | DDPG | ComSD |
|---|---|---|---|---|---|---|---|---|---|
| Walker | 13.47 $\pm$ 0.06 | 22.28 $\pm$ 0.08 | 7.34 $\pm$ 0.18 | 18.13 $\pm$ 4.65 | 11.02 $\pm$ 0 | 5.19 $\pm$ 0.14 | 7.34 $\pm$ 0.14 | 4.34 $\pm$ 0.03 | 7.5 $\pm$ 0.03 |
| Quadruped | 13.62 $\pm$ 0.1 | 23.01 $\pm$ 0.0 | 7.6 $\pm$ 0.21 | 13.42 $\pm$ 2.72 | 11.18 $\pm$ 0.01 | 5.45 $\pm$ 0.02 | 6.96 $\pm$ 1.37 | 4.49 $\pm$ 0.05 | 7.74 $\pm$ 0.03 |
| Jaco | 13.72 $\pm$ 0.03 | 22.76 $\pm$ 0.07 | 7.61 $\pm$ 0.03 | 14.88 $\pm$ 3.11 | 11.39 $\pm$ 0.03 | 6.4 $\pm$ 1.09 | 8.11 $\pm$ 0.02 | 4.78 $\pm$ 0.1 | 7.91 $\pm$ 0.02 |

Table 11: **Fine-tuning time (hours with decimal minutes) comparison across baselines.** Results are computed over 10 random seeds and reported as mean $\pm$ standard deviation.

| Domain | AMPED (Ours) | CeSD | CIC | BeCL | APT | RND | DIAYN | ComSD |
|---|---|---|---|---|---|---|---|---|
| Walker | 0.35 $\pm$ 0 | 0.7 $\pm$ 0 | 0.26 $\pm$ 0 | 0.24 $\pm$ 0 | 0.23 $\pm$ 0 | 0.36 $\pm$ 0 | 0.37 $\pm$ 0.01 | 0.25 $\pm$ 0 |
| Quadruped | 0.35 $\pm$ 0.01 | 0.73 $\pm$ 0.01 | 0.29 $\pm$ 0.01 | 0.29 $\pm$ 0.01 | 0.26 $\pm$ 0.01 | 0.44 $\pm$ 0.01 | 0.4 $\pm$ 0.01 | 0.28 $\pm$ 0.01 |
| Jaco | 0.32 $\pm$ 0.03 | 0.72 $\pm$ 0.02 | 0.27 $\pm$ 0.01 | 0.29 $\pm$ 0 | 0.26 $\pm$ 0.02 | 0.28 $\pm$ 0.06 | 0.38 $\pm$ 0 | 0.26 $\pm$ 0 |

### I.8 REPRODUCING BASELINES

All baseline methods were integrated from their respective open-source implementations and evaluated under our unified settings. In the Maze environment, DIAYN, CIC, and BeCL were reproduced using the EDL repository (Campos, 2020), and ComSD was imported from its official codebase (Liu, 2025)). Since no public implementation of CeSD exists for the Maze tasks, we followed the visualizations described in the original CeSD paper.

For the URLB, we leveraged the official URLB code (Laskin & Yarats, 2025) to reproduce DIAYN, RND, and APT. CIC, BeCL, and CeSD were run using their respective public implementations (Laskin & Liu, 2022; Yang, 2023; Bai & Yang, 2024). As ComSD lacks an official URLB release, we reimplemented it from its 2D-Maze variant, strictly adhering to the hyperparameters reported in its original publication.

Using the unmodified hyperparameters provided in the official CeSD codebase and paper, we were unable to reproduce the authors' reported performance. Our analysis indicates that CeSD exhibits substantially higher variance across random seeds than competing methods. Although minor implementation or evaluation differences cannot be entirely ruled out, the magnitude of the variance suggests that the observed gap is primarily attributable to CeSD's inherent instability rather than to specific experimental deviations (Table 13).

### I.9 EXPERIMENTAL SETUP AND REPRODUCIBILITY

All experiments were conducted on a Windows 11 workstation equipped with an AMD Ryzen 7 7700 8-core processor (3.80 GHz), 64GB DDR5 RAM, and an NVIDIA RTX 3060 GPU (12GB GDDR6).

Each experiment was run on a single GPU. The detailed wall-clock time for training and fine-tuning are summarized in Table 10 and Table 11.

We implemented all experiments in Python 3.8.10, using PyTorch (v1.9.0+cu111) as the primary deep learning framework. The DeepMind Control suite (Tassa et al., 2018) (dm-control v1.0.8) was used for environment simulation, and agent-environment communication was handled through the dm_env interface (v1.6).

## J  COMPARISON WITH METRA

We compare AMPED with METRA (Park et al., 2024), a metric-aware skill-based reinforcement learning method designed primarily for high-dimensional environments. Table 12 reports results on URLB. Because METRA is not available in URLB, we re-implemented it following the original paper, evaluating two variants: (i) *Discrete METRA*, which uses a categorical skill variable, and (ii) *Continuous METRA*, which samples skills from a Gaussian distribution.

In our experiments, both METRA variants underperform AMPED across all evaluated tasks, despite METRA outperforming CIC in the high-dimensional pixel-based settings reported in Park et al. (2024). This discrepancy highlights that the comparison is not entirely symmetric: METRA was developed for high-dimensional environments, where exploration is particularly challenging, whereas our evaluation focuses on the URLB control suite to isolate and validate the effect of balancing diversity and exploration on downstream task performance. Extending our framework to pixel-based domains is a natural next step, and investigating how METRA's metric-aware abstraction interacts with explicit exploration-diversity balancing in high-dimensional state spaces remains an interesting direction for future work.

Table 12: **Performance comparison between AMPED and METRA.** All results are computed using 10 seeds. Across all three environments, both discrete and continuous METRA variants underperform compared to AMPED.

| Domain | Task | AMPED (Ours) | Discrete Metra | Continuous Metra |
|---|---|---|---|---|
| Walker | Flip | $674 \pm 105$ | $329 \pm 50$ | $373 \pm 64$ |
| | Run | $467 \pm 103$ | $174 \pm 36$ | $274 \pm 48$ |
| | Stand | $951 \pm 38$ | $764 \pm 138$ | $781 \pm 154$ |
| | Walk | $929 \pm 19$ | $406 \pm 95$ | $502 \pm 206$ |
| | Sum | 3021 | 1673 | 1930 |
| Quadruped | Jump | $720 \pm 32$ | $218 \pm 194$ | $183 \pm 148$ |
| | Run | $494 \pm 53$ | $144 \pm 46$ | $127 \pm 86$ |
| | Stand | $906 \pm 67$ | $333 \pm 122$ | $240 \pm 127$ |
| | Walk | $890 \pm 59$ | $153 \pm 65$ | $160 \pm 117$ |
| | Sum | 3010 | 849 | 710 |
| Jaco | Re. bottom left | $143 \pm 32$ | $14 \pm 12$ | $26 \pm 10$ |
| | Re. bottom right | $144 \pm 25$ | $11 \pm 12$ | $21 \pm 29$ |
| | Re. top left | $140 \pm 39$ | $9 \pm 11$ | $21 \pm 14$ |
| | Re. top right | $154 \pm 46$ | $5 \pm 7$ | $24 \pm 21$ |
| | Sum | 581 | 39 | 93 |

## K  LIMITATIONS

As with any research, our approach presents several limitations that highlight opportunities for future investigation:

**Better gradient conflict resolver.** Although PCGrad is easy to implement and powerful, it has a few limitations. First, Liu et al. (2021) demonstrates that PCGrad does not preserve the original

objectives; instead, it merely guarantees convergence to the Pareto set. More advanced gradient conflict-resolution techniques have since been developed; future work can select the method best suited for SBRL.

**Removing heuristics.** Although we have introduced theory-based gradient surgery to balance exploration and diversity of skills, we still use the rule of thumb such as positive hyperparameters $\alpha, \beta$ for $r_{\text{total}} = r_{\text{diversity}} + \alpha r_{\text{entropy}} + \beta r_{\text{rnd}}$. Future work should eliminate such empirical rule of thumb.

**Inaccuracy and inefficiency of Estimators.** AnInfoNCE lacks precision, so future research should consider approaches that tighten the MI bound. And for entropy which has a high computational overhead, one should explore methods that are computationally efficient and capable of functioning effectively in high-dimensional state spaces. Model-based approaches, such as those using normalized flows (Ao & Li, 2022), could be potential solutions.

**Better objectives.** The diversity term adopted from the BeCL paper influences entropy, leading to gradient conflicts. Future research could focus on developing better objectives that maintain diversity without compromising entropy. In addition to entropy and RND based exploration, there has been a lot of research going on (Ladosz et al., 2022). One may find a better way to explore more efficiently and effectively.

**Balancing Other Factors Beyond Diversity and Exploration.** While our work primarily focuses on diversity and exploration, other aspects are also being actively studied to improve performance. Exploring how to harmonize our method with these additional aspects could be a valuable direction for future research. For instance, recent studies, such as (Park et al., 2023), rewards states that are difficult to reach.

**Fixed Number of Skills.** The current model treats the number of skills as a fixed hyperparameters, which is of course not ideal across different environments; see Figure C. While too few skills limit overall state coverage, once exploration saturates, adding more skills offers no further benefit. Developing mechanisms to dynamically adjust the number of skills according to the environment's requirements could enhance flexibility and performance.

**Relation Between Gradient Conflict and Environment.** Our experiments indicate that the degree of gradient conflict is strongly task-dependent. For example, Figure 5 shows that the percentage of conflicting gradients is noticeably lower in the Jaco domain than in Walker or Quadruped. This suggests that in some environments, behaviors that improve exploration (e.g., moving in varied directions) might naturally also promote diversity, leading to more aligned gradients; whereas in others exploratory behavior may cause different skills to visit overlapping regions, increasing conflict with the diversity objective. A more systematic characterization of how environment structure influences exploration-diversity gradient interactions, and how to adapt AMPED to these regimes, is an interesting direction for future work.

## L  LLM USAGE

We used a large language model (LLM) solely for language editing. Concretely, the LLM assisted with grammar and style polishing, LaTeX phrasing (e.g., equation and caption wording), and improving clarity and concision of author-written text. The LLM was not used to generate ideas, design algorithms, select hyperparameters, run experiments, analyze data, create figures/tables, write code, or produce mathematical results.

# M    VISUALIZATION OF SKILLS

Figure 17 illustrates the skills acquired during the pretraining stage for each environment.

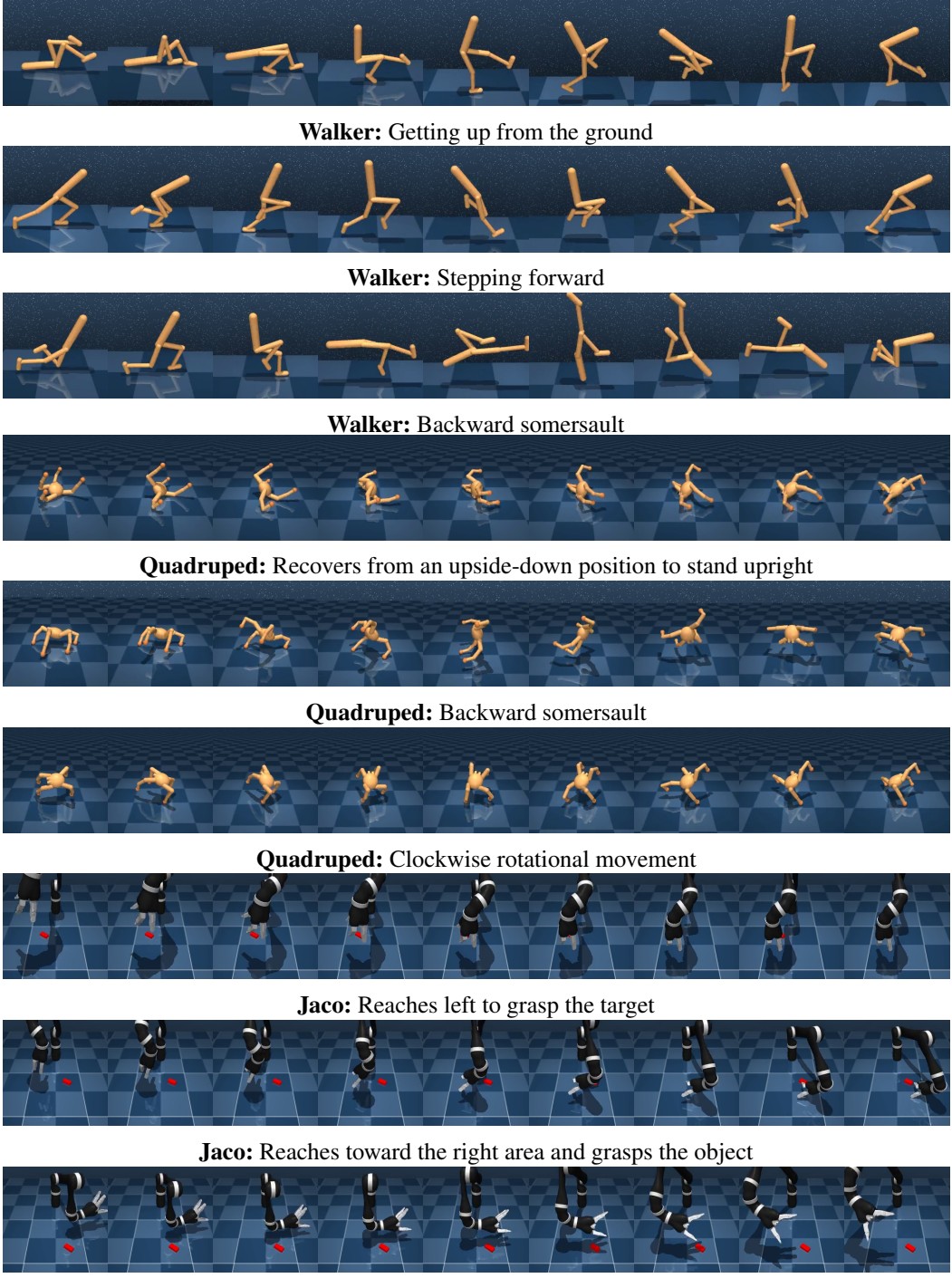

**Walker:** Getting up from the ground

**Walker:** Stepping forward

**Walker:** Backward somersault

**Quadruped:** Recovers from an upside-down position to stand upright

**Quadruped:** Backward somersault

**Quadruped:** Clockwise rotational movement

**Jaco:** Reaches left to grasp the target

**Jaco:** Reaches toward the right area and grasps the object

**Jaco:** Upward lifting motion while attempting to grasp the target

Figure 17: **Representative skills learned by our method. Walker** skills include rising from a supine position, stepping forward, and performing a backward somersault. **Quadruped** skills demonstrate self-righting, acrobatic flips, and rotational maneuvers. **Jaco** skills capture precise arm motions such as leftward reaching, rightward grasping, and upward lifting toward a target.

## N    NUMERICAL RESULTS

### N.1    PER-TASK EPISODE RETURNS ON URLB DOMAINS

Table 13: **Performance comparison with baselines.** Numerical results corresponding to Figure 8. Results are computed over 10 random seeds and reported as mean $\pm$ standard deviation. The best result is shown in **bold**, and the second-best is underlined.

| Domain | Task | AMPED (Ours) | CeSD | CIC | BeCL | APT | RND | DIAYN | DDPG | ComSD |
|---|---|---|---|---|---|---|---|---|---|---|
| Walker | Flip | $\underline{686} \pm 133$ | $623 \pm 90$ | $637 \pm 108$ | $625 \pm 66$ | $\mathbf{729} \pm 129$ | $483 \pm 71$ | $329 \pm 39$ | $531 \pm 46$ | $488 \pm 57$ |
| | Run | $\underline{517} \pm 49$ | $377 \pm 89$ | $454 \pm 82$ | $435 \pm 73$ | $\mathbf{542} \pm 73$ | $371 \pm 86$ | $183 \pm 35$ | $327 \pm 115$ | $341 \pm 100$ |
| | Stand | $947 \pm 19$ | $915 \pm 68$ | $939 \pm 33$ | $\mathbf{953} \pm 11$ | $\underline{949} \pm 20$ | $892 \pm 47$ | $716 \pm 127$ | $905 \pm 56$ | $937 \pm 17$ |
| | Walk | $\underline{886} \pm 63$ | $805 \pm 133$ | $874 \pm 67$ | $818 \pm 189$ | $\mathbf{892} \pm 62$ | $792 \pm 139$ | $434 \pm 94$ | $736 \pm 149$ | $826 \pm 111$ |
| | sum | $\underline{3036}$ | $2720$ | $2904$ | $2831$ | $\mathbf{3112}$ | $2538$ | $1662$ | $2499$ | $2592$ |
| Quadruped | Jump | $\underline{699} \pm 68$ | $529 \pm 160$ | $580 \pm 120$ | $668 \pm 44$ | $\mathbf{720} \pm 32$ | $643 \pm 50$ | $555 \pm 159$ | $337 \pm 129$ | $607 \pm 101$ |
| | Run | $\mathbf{493} \pm 54$ | $390 \pm 212$ | $442 \pm 72$ | $394 \pm 98$ | $\underline{468} \pm 97$ | $435 \pm 34$ | $398 \pm 88$ | $251 \pm 112$ | $336 \pm 91$ |
| | Stand | $816 \pm 150$ | $\mathbf{853} \pm 40$ | $693 \pm 193$ | $640 \pm 215$ | $821 \pm 192$ | $\underline{839} \pm 45$ | $644 \pm 179$ | $511 \pm 253$ | $684 \pm 201$ |
| | Walk | $\mathbf{816} \pm 116$ | $562 \pm 322$ | $630 \pm 183$ | $635 \pm 205$ | $\underline{758} \pm 192$ | $571 \pm 90$ | $404 \pm 200$ | $209 \pm 60$ | $396 \pm 182$ |
| | sum | $\mathbf{2824}$ | $2334$ | $2345$ | $2337$ | $\underline{2767}$ | $2488$ | $2001$ | $1308$ | $2023$ |
| Jaco | Re. bottom left | $\underline{139} \pm 34$ | $136 \pm 25$ | $135 \pm 19$ | $\mathbf{148} \pm 26$ | $120 \pm 24$ | $101 \pm 24$ | $20 \pm 21$ | $133 \pm 57$ | $126 \pm 24$ |
| | Re. bottom right | $\underline{140} \pm 21$ | $134 \pm 7$ | $\mathbf{152} \pm 23$ | $\underline{140} \pm 22$ | $126 \pm 25$ | $115 \pm 24$ | $22 \pm 20$ | $115 \pm 62$ | $111 \pm 41$ |
| | Re. top left | $130 \pm 38$ | $\mathbf{175} \pm 8$ | $\underline{137} \pm 21$ | $123 \pm 35$ | $124 \pm 22$ | $97 \pm 30$ | $22 \pm 22$ | $101 \pm 60$ | $126 \pm 25$ |
| | Re. top right | $\underline{146} \pm 49$ | $97 \pm 29$ | $\mathbf{149} \pm 19$ | $116 \pm 31$ | $113 \pm 25$ | $122 \pm 30$ | $12 \pm 12$ | $87 \pm 53$ | $121 \pm 15$ |
| | sum | $\underline{555}$ | $542$ | $\mathbf{573}$ | $527$ | $483$ | $435$ | $76$ | $436$ | $484$ |
| **Total** | sum | $\mathbf{6415}$ | $5596$ | $5822$ | $5705$ | $\underline{6362}$ | $5461$ | $3747$ | $4243$ | $5099$ |

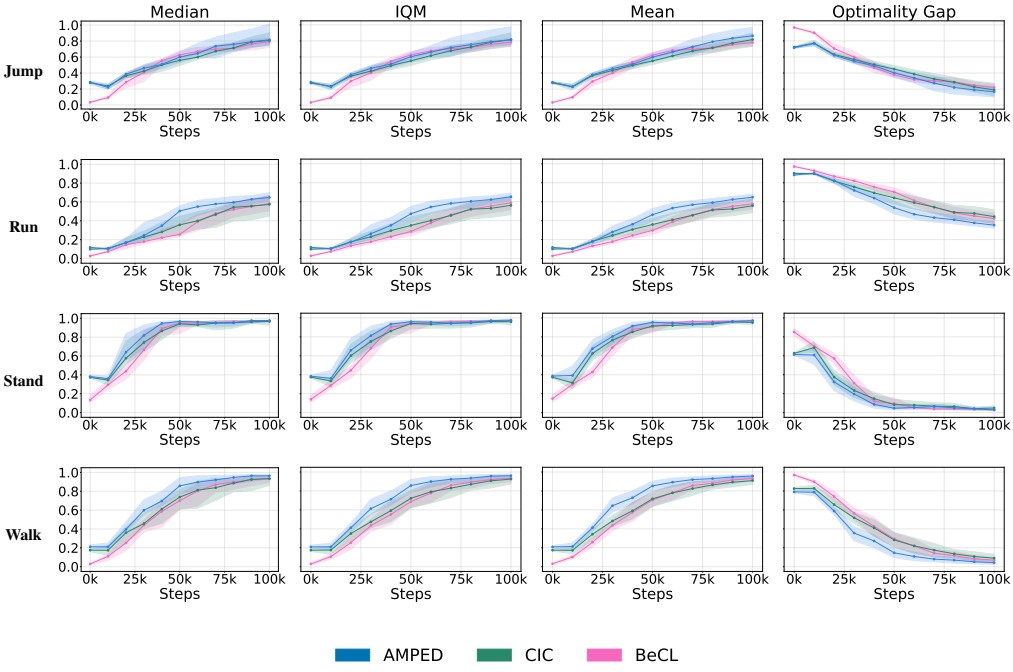

Figure 18: **Performance on Walker tasks normalized by experts' scores.** Results show Median, IQM, Mean, and Optimality Gap computed over 10 random seeds. Shaded regions represent 95% confidence intervals.

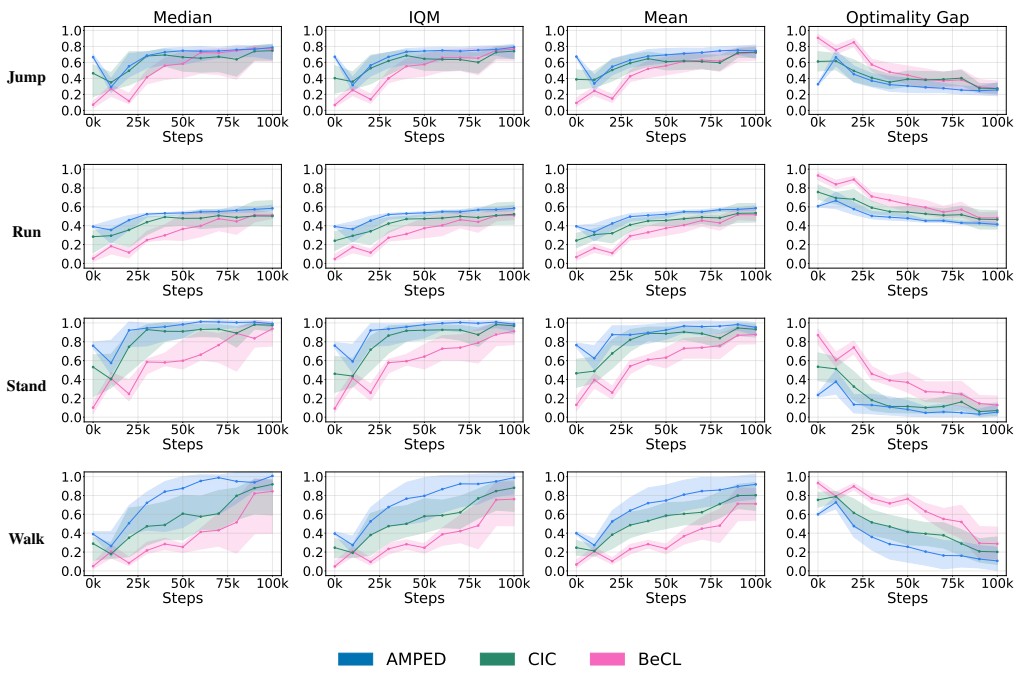

Figure 19: **Performance on Quadruped tasks normalized by experts' scores.** Results show Median, IQM, Mean, and Optimality Gap computed over 10 random seeds. Shaded regions represent 95% confidence intervals.

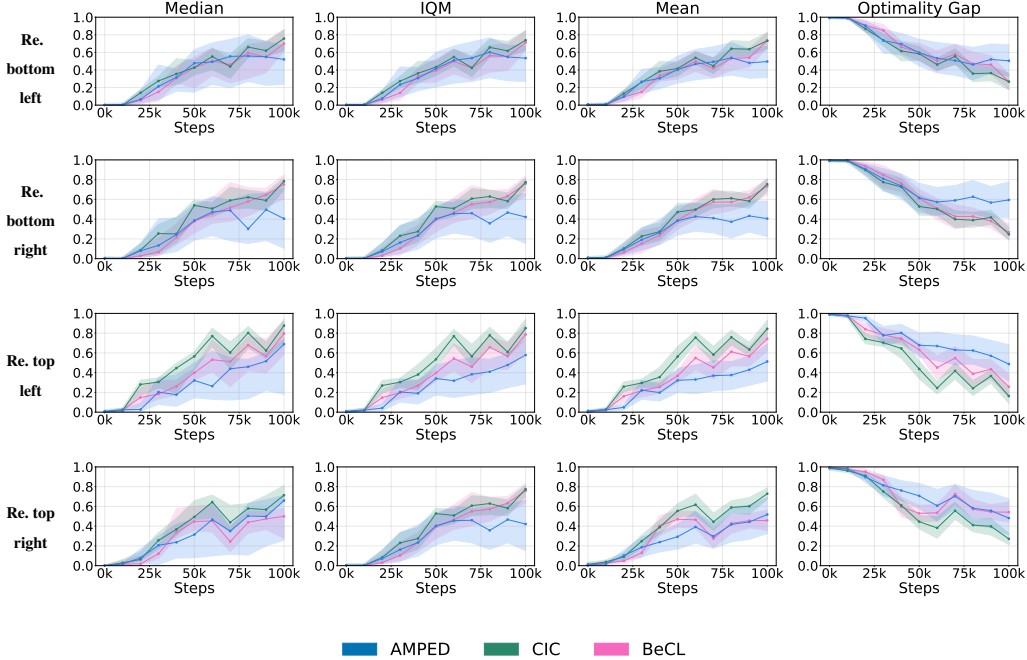

Figure 20: **Performance on Jaco tasks normalized by experts' scores.** Results show Median, IQM, Mean, and Optimality Gap computed over 10 random seeds. Shaded regions represent 95% confidence intervals.

We report per-task episode returns (mean $\pm$ standard devition over 10 seeds) following the evaluation protocol of Agarwal et al. (2021). All methods are pretrained for 2M steps with only intrinsic rewards,

then finetuned for 100k steps on each downstream task by adding the extrinsic reward. Table 13 presents results on the Walker, Quadruped, and Jaco domains.

AMPED achieves at least second best performance in almost every discipline, and has the highest total sum, which confirms that our method consistently perfoms well on different tasks (Table 13). More concretely, AMPED achieves a cumulative total sum of 6415, which is the highest among all methods. The next best is APT with 6362, trailing AMPED by 53 points, and CIC comes third at 5822, well behind by 593 points. These per-task breakdowns confirm that AMPED's joint handling of entropy, RND, and diversity objectives delivers consistently strong performance across a diverse set of URLB challenges. Although CeSD and ComSD also aim to balance diversity and exploration, AMPED outperforms both on all but one task (Re., top left), demonstrating that our unified objective formulation is more effective.

We additionally include time-series plots of downstream fine-tuning returns for AMPED, CIC, and BeCL (Figure 18, Figure 19, Figure 20). In both Walker and Quadruped, AMPED consistently surpasses CIC and BeCL throughout training, whereas on Jaco its performance is comparable to the two methods, which focus exclusively on exploration or diversity. These results underscore the importance of jointly balancing both objectives.

## N.2    ABLATION STUDY ON PROJECTION RATIO

As detailed in Section 4.3, AMPED's balanced projection ratio mitigates gradient conflicts and improves skill learning across diverse environments. The full numerical results corresponding to Figure 9 are provided in Table 14. The projection ratios used for AMPED (Ours) are listed in Appendix I.5.

Table 14: **Performance under different projection-ratio settings** ($p$). Numerical results corresponding to Figure 9. **AMPED (Ours)** uses the default projection ratio, chosen separately for each environment. Results are computed over three random seeds and reported as mean $\pm$ standard deviation, except AMPED, which uses 10 seeds. The best result is shown in **bold**, and the second-best is underlined.

| Domain | Task | AMPED (Ours) | $p = 0.0$ | $p = 1.0$ |
|---|---|---|---|---|
| Walker | Flip | **674** $\pm$ 105 | 628 $\pm$ 55 | 606 $\pm$ 37 |
| | Run | 467 $\pm$ 103 | 427 $\pm$ 44 | **533** $\pm$ 71 |
| | Stand | **951** $\pm$ 38 | 949 $\pm$ 3 | 931 $\pm$ 12 |
| | Walk | 929 $\pm$ 19 | **939** $\pm$ 13 | 828 $\pm$ 73 |
| | Sum | **3021** | 2943 | 2898 |
| Quadruped | Jump | **720** $\pm$ 34 | 706 $\pm$ 19 | 661 $\pm$ 34 |
| | Run | 494 $\pm$ 56 | 483 $\pm$ 6 | **501** $\pm$ 29 |
| | Stand | **906** $\pm$ 71 | 859 $\pm$ 109 | 905 $\pm$ 4 |
| | Walk | **890** $\pm$ 62 | 613 $\pm$ 183 | 626 $\pm$ 129 |
| | Sum | **3010** | 2661 | 2693 |
| Jaco | Re. bottom left | **143** $\pm$ 34 | 133 $\pm$ 44 | 126 $\pm$ 10 |
| | Re. bottom right | **144** $\pm$ 27 | 108 $\pm$ 53 | 140 $\pm$ 17 |
| | Re. top left | **140** $\pm$ 41 | 84 $\pm$ 37 | 126 $\pm$ 29 |
| | Re. top right | **154** $\pm$ 49 | 103 $\pm$ 32 | 139 $\pm$ 34 |
| | Sum | **581** | 428 | 531 |

## N.3 ABLATION STUDY ON NUMBER OF SKILLS

As discussed in Section 4.3, the number of skills plays an important role in downstream performance. The full numerical results corresponding to Figure 10 are summarized in Table 15. AMPED (Ours) uses 16 skills in all domains.

Table 15: **Performance under different skill dimension setting.** Results are computed over three random seeds and reported as mean ± standard deviation, except AMPED, which uses 10 seeds.

| Domain | Task | **AMPED (Ours)** | Skill dimension 8 | Skill dimension 32 |
|---|---|---|---|---|
| Walker | Flip | $\underline{674} \pm 105$ | $552 \pm 86$ | $\mathbf{680} \pm 196$ |
| | Run | $467 \pm 103$ | $\underline{468} \pm 84$ | $\mathbf{511} \pm 32$ |
| | Stand | $\mathbf{951} \pm 38$ | $\underline{942} \pm 13$ | $907 \pm 83$ |
| | Walk | $\underline{929} \pm 19$ | $929 \pm 25$ | $\mathbf{940} \pm 22$ |
| | Sum | $\underline{3021}$ | $2891$ | $\mathbf{3038}$ |
| Quadruped | Jump | $\mathbf{720} \pm 32$ | $\underline{493} \pm 182$ | $386 \pm 116$ |
| | Run | $\mathbf{494} \pm 53$ | $\underline{380} \pm 73$ | $247 \pm 49$ |
| | Stand | $\mathbf{906} \pm 67$ | $\underline{705} \pm 202$ | $602 \pm 203$ |
| | Walk | $\mathbf{890} \pm 59$ | $\underline{488} \pm 201$ | $388 \pm 190$ |
| | Sum | $\mathbf{3010}$ | $\underline{2066}$ | $1623$ |
| Jaco | Re. bottom left | $\mathbf{143} \pm 32$ | $137 \pm 38$ | $\underline{142} \pm 13$ |
| | Re. bottom right | $\underline{144} \pm 25$ | $\mathbf{153} \pm 22$ | $124 \pm 15$ |
| | Re. top left | $\mathbf{140} \pm 39$ | $\underline{117} \pm 49$ | $108 \pm 32$ |
| | Re. top right | $\mathbf{154} \pm 46$ | $\underline{148} \pm 15$ | $132 \pm 44$ |
| | Sum | $\mathbf{581}$ | $\underline{555}$ | $506$ |

