# OpenReview forum: "AMPED: Adaptive Multi-objective Projection for balancing Exploration and skill Diversification"
_ICLR.cc/2026/Conference — ICLR 2026 Poster_

### Official Review · Reviewer_mrH4 · 2025-10-20

**Soundness:** 3
**Presentation:** 3
**Contribution:** 3
**Rating:** 6
**Confidence:** 4

**Summary:**

This paper introduces "AMPED", a method for unsupervised skill learning and efficient adaptation to downstream tasks.

**Main contribution:** During pretraining, the method explicitly handles two potentially conflicting objectives:
1. Achieving wide exploration of the state space
2. Achieving a diverse distribution of skills

The pretraining method relies on gradient surgery to ensure that conflicts between the gradients for the two objectives are resolved, thereby avoiding that the competing objectives negatively affect the pretraining process.

After pretraining, the downstream task can efficiently be solved by exploiting the skills discovered during pretraining.
For this, AMPED learns a selector over the skills to maximize the downstream reward, which allows for more efficient adaptation compared to randomly sampling or fine-tuning skills.

The method is evaluated against a large number of unsupervised skill discovery baselines, on both didactic maze environments and more challenging continuous control tasks, all showing strong performance gains of AMPED compared to prior works.
An ablation over AMPED's components helps understanding the importance of each.

**Strengths:**

- The motivation is clear: conflicting objectives can deteriorate performance during pretraining.
- The paper is well written and easy to follow. Key mathematical objects and issues are presented and explained clearly.
- The paper formally connects the diversity of the pretrained skills with the number of samples needed to discover the optimal skill during the downstream adaptation step.
- The experiments include a wide range of representative baselines, and the evaluation environments feature both a didactic 2D maze example and challenging URLB environments.
- The results are strong, convincing, and consistent with the central claims of the paper.

**Weaknesses:**

- The key insights are already well understood, and the novelty is somewhat limited. The paper’s key insight, line 053, that conflicts can arise between gradients for different objectives, is well understood in the multi-objective RL and optimization literature, and thus not a strong novel finding.

- To the best of my knowledge, the paper does not introduce fundamentally new approaches to solving the challenges in multi-objective optimization, but rather combines existing techniques and applies them to unsupervised skill discovery and downstream adaptation. The contribution is still valuable, but the paper is relatively applied in nature, IMO.
- Without digging into the related works, it is unclear whether estimating the mutual information with AInfoNCE is a contribution of the authors. If so, I acknowledge it as a strength, slightly weakening my limited-novelty critique.
- The learned hierarchical selector during fine-tuning is not conceptually different from prior works such as DIAYN, which also features a hierarchical selector over discovered skills.

**Clarity:** Lines 300–304 do not sufficiently explain how the skill selector is learned exactly, and what skill fine-tuning entails. Please elaborate and explain this more clearly.
  - Is “fine-tuning” used interchangeably with learning the skill selector?
  - Are the skills themselves being fine-tuned to maximize the downstream reward?
  This needs to be made more explicit. Consider describing clearly which parameters of your architecture are updated during the adaptive skill selection step, and how.

**Nitpicks:**
- Lines 183–184: $z$ is defined as a skill, but $z^\ast$ is defined as a best policy, which is confusing. Does $z^\ast$ always induce the optimal state visitation distribution? What if that \$z^\ast$ is given to a suboptimal policy that cannot execute the optimal skill well?
- Lines 251–252: I assume (1) and (2) are indices of different skills? The wording can be understood like they belong to the same skill.
- Line 86 mentions “statistically significant results,” but no formal hypothesis test ($p < 0.05$) is performed. Consider changing the wording to “consistent improvements” or similar.

**Questions:**

As part of the ablation study, lines 425–431 discuss using different projection ratios. Can you elaborate a bit more on what is actually meant by *AMPED (ours)* here? Is it the procedure outlined in lines 287–295?
  If yes, I think it would improve clarity to make that connection more explicit in the main text.

---

> ### Author Response · Authors · 2025-11-25
>
> We appreciate the reviewer’s detailed critique and constructive guidance. The feedback substantially strengthened the clarity and presentation of our work. We address each point below.
>
> ### 1. Clarifying Our Contributions
>
> We understand the reviewer’s concern about novelty. While AMPED does build on existing ideas, our contributions extend beyond a simple combination of components:
>
> - **Balancing exploration and diversity:** We provide empirical evidence that jointly handling both objectives is critical in SBRL, a dimension underexplored in prior work.
> - **Gradient-conflict perspective:** We identify and measure conflict between exploration and diversity objectives, and adapt PCGrad with a probabilistic projection mechanism tailored to this setting.
> - **Exploration design:** We show that combining RND with a particle-based entropy estimator, though not previously explored in skill-based RL, significantly strengthens exploration.
> - **MI estimation for skills:** We introduce AnInfoNCE as an MI estimator for skill diversification. Although AnInfoNCE is an existing method, it has not been applied to skill-based RL before, and our ablations demonstrate clear benefits.
> - **Theory:** We provide a theoretical analysis highlighting the role of diversity in skill-based RL, an aspect less theoretically grounded than exploration in prior literature.
>
> Together, these components offer a coherent framework that consistently improves skill-based RL performance across tasks.
>
> ### 2. Clarifications
>
> > Lines 300–304 do not sufficiently explain how the skill selector is learned exactly, and what skill fine-tuning entails. Please elaborate and explain this more clearly.
>
> * At each time step, the selector samples a skill according to $p(z \mid s)$. The policy conditioned on the selected skill is then updated using the downstream task reward. The skill selector is trained to maximize the same reward signal as the policy, and both the policy parameters and selector parameters are updated at every step. We have clarified this in the paper.
>
> > Lines 183–184: $z$ is defined as a skill, but $z^\ast$ is defined as a best policy, which is confusing. Does $z^\ast$ always induce the optimal state visitation distribution? What if that $z^\ast$ is given to a suboptimal policy that cannot execute the optimal skill well?
>
> * Thank you for catching this confusion. This is a notation issue: $z^\ast$ is meant to be a best skill. It will not induce the globally optimal policy. Rather, it induces the policy in our skill set whose state distribution is closest to that of the target policy. The notion of “closest” is defined in terms of the distance between induced state distributions.
>
> > Lines 251–252: I assume (1) and (2) are indices of different skills? The wording can be understood like they belong to the same skill.
>
> * The reviewer is correct that (1) and (2) represent two samples from the same skill. Maximizing $I(S^{(1)}, S^{(2)})$ clusters states generated by the same skill while separating different skills. We have clarified this wording.
>
> > Line 86 mentions "statistically significant results," but no formal hypothesis test ($p < 0.05$) is performed. Consider changing the wording to “consistent improvements” or similar.
>
> * Following [1], we use 95% stratified bootstrap confidence intervals and robust metrics such as IQM and Optimality Gap, for which AMPED’s confidence intervals are clearly separated from those of baselines on most tasks. We replaced "statistically significant results" with "statistically well-supported and consistent improvements."
>
> > As part of the ablation study, lines 425–431 discuss using different projection ratios. Can you elaborate a bit more on what is actually meant by AMPED (ours) here? Is it the procedure outlined in lines 287–295? If yes, I think it would improve clarity to make that connection more explicit in the main text.
>
> * Yes, "AMPED (Ours)" refers to the procedure described in the section 4. We clarified this connection in the revised text.
>
> We thank the reviewer again for the thoughtful and constructive feedback. The revisions significantly improved both presentation and conceptual clarity. We hope the updated explanations address the concerns and will be helpful in the reviewer’s final evaluation.
>
> [1] Agarwal et al. Deep reinforcement learning at the edge of the statistical precipice.

---

> > ### Comment · Reviewer_mrH4 · 2025-11-26
> >
> > Thank you for your response and clarifications.
> >
> > The paper was in a good state before and my concerns were minor. Based on the additional clarifications in the revised manuscript, I am increasing my score to 8.

---

> ### Author Response · Authors · 2025-11-26
>
> Thank you for the thoughtful assessment and the updated score. We appreciate your careful reading and constructive feedback throughout the review process.

---

### Official Review · Reviewer_B5oH · 2025-10-31

**Soundness:** 3
**Presentation:** 2
**Contribution:** 2
**Rating:** 6
**Confidence:** 3

**Summary:**

This paper studies unsupervised skill learning and proposed AMPED, a novel method that aims to learn skills that are both diverse and explore the state space well. To do this, AMPED relies on an intrinsic reward based on an entropy bonus combined with RND for exploration, and an intrinsic reward based on contrastive learning for incentivizing the skills to be diverse. Finally, AMPED uses a gradient-surgery projection procedure to balance the exploration and diversity gradients and avoid gradient conflict. On the URLB benchmark, they find their method to outperform all baselines in terms of mean, IQM, and optimality gap. The authors also provide ablation studies to test the importance of the entropy bonuses, the RND bonus, the Anisotropic InfoNCE reward, the gradient surgery procedure, and the skill selector.

**Strengths:**

1. Evaluation was done over 10 random seeds and reports results according to recommendations from the Rliable framework.
2. Strong results: outperforms all baselines in terms of mean, IQM, and optimality gap.
3. The motivation for having explicit exploration and diversity rewards is clear.
4. Figure 4 provides a very helpful visualization of the gradient surgery procedure.
5. Figure 5 provides a good qualitative visualization of the exploration performance and the distinguishabiltiy of the learned skills for all considered methods in a toy maze.

**Weaknesses:**

1. Part 2 of Figure 2 is nothing new: this seems to be a standard HRL setup for skill discovery methods? In general, the paper emphasizes the benefits of the skill selector, while to me this just seems like HRL over the learned skills which has been done in prior work already (see METRA or CSF). Could the authors comment on this? Is there any differences or contributions that I’m missing?
2. Page 4: “is the discounted total state distribution” → do the authors mean the discounted state occupancy measure here? If so, I’d recommend replacing it with that since it’s the canonical terminology for this.
3. Section 4.4: “… this indicates faster convergence of AMPED” → based on Table 3, I disagree with the authors on this claim. It appears to me all the numbers reported fall within each other’s error bars, questioning the statistical significance. Could the authors comment on this?

**Questions:**

1. Page 7: “In order to ensure a fair comparison, we fine-tuned all methods under identical conditions without using the skill selector.” → how does that work? Every skill gets fine-tuned towards the same task reward? Does this preserve the skill behavior? Do the skills change a lot during fine-tuning?
2. Section 3.1: "and ρ⋆ ∈∆(S^H) be a corresponding state distribution." -> If this is a state distribution, shouldn't it just be ∆(S) instead of ∆(S^H)? The latter seems like a distribution over trajectories. This also confuses me later where it says "Draw n i.i.d. trajectories from optimal policy S(1),...,S(n) ∼ρ⋆". This seems like drawing n states instead of n trajectories?
3. Could the authors expand a bit on exactly why it's necessary to have both RND and entropy exploration? There's some discussion about it on page 5 but it wasn't super clear to me.
4. Page 5: "However, if the supports of different skill coverages become disjoint, the diversity loss Ldiversity no longer enforces inter-skill distributional separation, which can lead to skill clustering rather than promoting broad coverage of the state space." -> Could the authors provide some clarification on this statement? I couldn't really follow what exactly is meant here.

---

> ### Author Response · Authors · 2025-11-25
>
> We thank the reviewer for the careful reading of our paper and the valuable insights. The suggestions have led to notable refinements, and we provide point-by-point replies below.
>
> ### 1. Skill Selector
>
> We agree with the reviewer that using a skill selector is common in hierarchical RL. However, many recent skill-based RL methods, including CeSD, BeCL, CIC, DIAYN, etc., do not include a dedicated skill-selection mechanism during downstream fine-tuning and instead rely on uniformly sampling skills at fixed intervals. To more effectively leverage the learned skills, we incorporate an adaptive skill selector. That said, we emphasize that our main contribution is not the skill selector itself, but the broader framework that jointly addresses the exploration-diversity trade-off.
>
> ### 2. Clarifications
> > Page 4: “is the discounted total state distribution” → do the authors mean the discounted state occupancy measure here? If so, I’d recommend replacing it with that since it’s the canonical terminology for this.
>
> * We have replaced “discounted total state distribution” with the canonical term discounted state occupancy measure, as suggested.
>
> > Section 4.4: “… this indicates faster convergence of AMPED” → based on Table 3, I disagree with the authors on this claim. It appears to me all the numbers reported fall within each other’s error bars, questioning the statistical significance. Could the authors comment on this?
>
> * We agree that the previous version of Section 4.4 did not align well with Theorem 1. After additional hypothesis testing, we found that two of the four tasks did not show statistically significant differences. More importantly, the experiment did not isolate the effect of diversity. Since modifying diversity also alters exploration and state coverage, empirical validation of the theorem is inherently difficult. Therefore, rather than attempting to empirically validate the theorem in Section 4.4, we removed that section and instead included a controlled analysis of the number of skills in Section 4.3.
>
> > Page 7: “In order to ensure a fair comparison, we fine-tuned all methods under identical conditions without using the skill selector.” → how does that work? Every skill gets fine-tuned towards the same task reward? Does this preserve the skill behavior? Do the skills change a lot during fine-tuning?
>
> * Following the protocol used in BeCL, CeSD, CIC, DIAYN, and other SBRL baselines, the skill is uniformly sampled from the entire skill set and switched every 50 steps.
>
> > Section 3.1: "and ρ⋆ ∈∆(S^H) be a corresponding state distribution." -> If this is a state distribution, shouldn't it just be ∆(S) instead of ∆(S^H)? The latter seems like a distribution over trajectories. This also confuses me later where it says "Draw n i.i.d. trajectories from optimal policy S(1),...,S(n) ∼ρ⋆". This seems like drawing n states instead of n trajectories?
>
> * You are correct. It should be $\rho$ $\in$ $\Delta(\mathcal{S})$, not $\Delta(\mathcal{S}^H)$. $\rho^\*$ is a state distribution, so the original phrasing “draw trajectories from $\rho^\*$” was misleading. We corrected these expressions accordingly.
>
> > Could the authors expand a bit on exactly why it's necessary to have both RND and entropy exploration? There's some discussion about it on page 5 but it wasn't super clear to me.
>
> * Entropy estimation via a particle-based entropy estimator requires O(nlogn) k-NN computations, where n denotes the number of states in the replay buffer, which is impractical for large buffers. Following APS, we truncate the buffer to 5,000 states, improving efficiency but degrading estimator fidelity. RND provides an inexpensive intrinsic reward (O(n)), but is noisy early in training when the buffer is small. Together, they complement each other and result in a more robust exploration mechanism. We clarified this reasoning in the revision.
>
> > Page 5: "However, if the supports of different skill coverages become disjoint, the diversity loss Ldiversity no longer enforces inter-skill distributional separation, which can lead to skill clustering rather than promoting broad coverage of the state space." -> Could the authors provide some clarification on this statement? I couldn't really follow what exactly is meant here.
>
> * Our intention was to state that CeSD’s diversity loss treats all non-overlapping trajectories equally and therefore does not distinguish between slightly separated and strongly separated skills. We revised the phrasing accordingly.
>
> We sincerely appreciate the reviewer’s comments, which have substantially strengthened the overall presentation of the paper. We believe our clarifications adequately address the raised questions, and we kindly ask the reviewer to consider these responses in their overall evaluation.

---

### Official Review · Reviewer_8RFz · 2025-10-31

**Soundness:** 3
**Presentation:** 3
**Contribution:** 3
**Rating:** 6
**Confidence:** 4

**Summary:**

This work proposes an adaptive training method to balance exploration and diversity in the application of unsupervised reinforcement learning (URL) for skill discovery. The authors argue that optimizing exploration and diversity in skill discovery tasks can be contradictory. Therefore, they design the algorithm from a multi-objective optimization perspective, defining "enhancing skill exploration" and "improving skill diversity" as two distinct objectives for the URL algorithm. They then propose using a gradient clipping approach to balance these two optimization goals. The proposed method demonstrates superior performance compared to existing baselines across multiple benchmarks. Furthermore, the paper includes comprehensive ablation studies that experimentally validate the necessity and correctness of the various components within the proposed algorithm.

**Strengths:**

- This paper is easy to follow.
- The results of the ablation study are comprehensive.

**Weaknesses:**

- Some content is unclearly explained, such as the specific form of $\rho$ and the meaning of the several metrics in Figure 6.
- The proof of Theorem 1 is currently difficult to ascertain as valid (see Questions for details).
- The experimental settings are too simple, primarily consisting of basic maze environments.
- The presentation of the conflicts during the skill learning process is not intuitive.

**Questions:**

- In the proof of Theorem 1, the authors state that a larger $\delta$ increases $\Delta$ and thus reduces the required sample size. However, Inequality 3 only provides a lower bound for $n$ and does not explicitly demonstrate that increased diversity reduces the necessary sample size. Could the authors clarify this point?
- What are the exact calculation of median, IQM (is it Interquartile Mean?), and mean scores and optimality gap in Figure6?
- Table 1 shows the percentage of conflicts. What does the corresponding learning curve look like for these conflicts?
- How does the dimension of $z$ affect the final policy performance?
- The presentation of the conflicts during the skill learning process is not intuitive. Could the authors make it clearer, perhaps by reporting the curve of gradient similarity changes?
- Is the gradient similarity related to the type of task? For instance, in some scenarios, the conflict between exploration and diversity might be small, while in others it could be large.

---

> ### Author Response · Authors · 2025-11-25
>
> We deeply appreciate the reviewer’s thorough assessment and helpful recommendations. The raised points significantly strengthened the revised version, and we address each of them individually in the following sections.
>
> ### 1. Clarifications
> > Some content is unclearly explained, such as the specific form of $\rho$ and the meaning of the several metrics in Figure 6.
>
> - We now clearly define $\rho$ as the discounted state-occupancy measure and expand the explanations of the evaluation metrics used in Figure 6, including IQM, and the optimality gap, the latter of which is not commonly used.
>
> > In the proof of Theorem 1, the authors state that a larger $\delta$ increases $\Delta$ and thus reduces the required sample size. However, Inequality 3 only provides a lower bound for n and does not explicitly demonstrate that increased diversity reduces the necessary sample size. Could the authors clarify this point?
>
> - Regarding Theorem 1, $\delta$ denotes the minimum pairwise distance between skills and serves as our diversity measure. Inequality (3) provides a lower bound on the required number of samples. Under our assumptions, this lower bound is a monotonically decreasing function of $\delta$. While this does not establish a full sample-complexity guarantee, it formally shows that lower diversity increases the lower bound, and intuitively, greater separation between skills makes them easier to distinguish from trajectory samples.
>
> ### 2. Gradient Conflict
>
> To make the conflict presentation more intuitive, we added gradient conflict ratio curves (Figure 5) and gradient similarity curves (Figure 6). These plots show that, although the magnitude varies across tasks, substantial gradient conflict consistently arises between the exploration and diversity objectives in all environments. Moreover, the trends are task-dependent: some tasks exhibit only mild conflict (i.e., small negative inner products), whereas others display substantially larger negative inner-product scales.
>
> ### 3. Number of Skills
> We included a controlled analysis on the number of skills in Section 4.3 (Q3). This ablation shows that diversity does not increase monotonically with the number of skills and that performance degrades when the skill set is either too small or too large. Intuitively, a small skill set may already provide sufficient coverage of the environment, leaving little opportunity for additional skills to contribute further diversity. Conversely, even in larger environments, diversity is optimized through an approximate mutual-information objective rather than an exact estimator, so increasing the number of skills does not guarantee proportional gains in separation. These observations highlight that selecting an appropriate skill dimension is essential for maximizing diversity and achieving strong downstream performance. Across all three domains, 16 skills (our default) yield the most stable and competitive results.
>
> We are grateful for the reviewer’s comments, which significantly improved both the theoretical discussion and experimental scope of the paper. We believe our clarifications address the raised questions, and we would appreciate the reviewer’s consideration of these responses in their overall judgement.

---

### Official Review · Reviewer_3YZa · 2025-11-01

**Soundness:** 3
**Presentation:** 3
**Contribution:** 2
**Rating:** 4
**Confidence:** 4

**Summary:**

This paper introduces a new framework for skill-based reinforcement learning called AMPED. During the pre-training phase, the method uses a "gradient-surgery projection" to mitigate conflicting gradient components. During the fine-tuning phase, it employs an "adaptive skill selector" to leverage the learned skills for downstream tasks. Experiments demonstrate that this method surpasses baselines on the URLB and Maze benchmarks, and proves that greater diversity.

**Strengths:**

1. The experimental analysis is thorough. In particular, extensive ablation studies are provided to demonstrate the effectiveness of each component in different scenarios.

2. The paper is exceptionally clear and logically fluent. The progression from the problem definition in the Introduction, to the Methods section, and then to the Experimental Analysis is easy to follow. Furthermore, the Appendix provides extremely detailed further analysis, enhancing comprehensibility.

3. The paper models this problem as a "gradient conflict" in multi-objective optimization and introduces "gradient-surgery projection" to resolve it. This is a very insightful and novel approach.

**Weaknesses:**

1. I find that the method presented in the paper largely consists of combining existing techniques, and it lacks significant methodological innovation.

2. Directly using SAC to learn a policy over the skill repertoire (i.e., changing the prior z) seems feasible. However, the paper does not couple the two RL training processes well, which may lead to unstable convergence and, as noted in Table 6, instances of ineffectiveness.

3. The experimental environments are relatively simple, with experiments primarily compared on the Maze and 12 URLB downstream tasks. I believe the contribution to practical applications (e.g., pixel-based environments) is minimal.

4. The algorithmic complexity is high (integrating CIC, RND, SAC, etc.), making the training process relatively heavy. Moreover, the efficiency analysis provided is not intuitive enough.

**Questions:**

1. The authors state that METRA was omitted because it does not exhibit performance improvements on the URLB relative to CeSD. However, I noted in the paper[1] that METRA outperforms CIC on some tasks. Given that CIC is a relatively strong performer in the authors' own experiments, could the authors please add supplementary experiments comparing against METRA?

2. Could you provide more comparisons and analysis of the algorithm's efficiency? For instance, by plotting learning curves that show the evaluation metrics as a function of training steps?

[1] METRA: SCALABLE UNSUPERVISED RL WITH METRIC-AWARE ABSTRACTION

---

> ### Author Response · Authors · 2025-11-25
>
> We are grateful for the reviewer’s insightful comments and careful evaluation of our manuscript. The feedback has meaningfully guided several improvements, and we provide detailed responses to questions and concerns below.
>
> ### 1. Comparison with METRA
>
> Following the reviewer’s suggestion, we carefully re-implemented METRA based on its original paper and official codebase, and added full comparisons on URLB. The implementation is included in the supplementary material, and results appear in Appendix J. Across all domains, AMPED consistently outperforms both the discrete and continuous versions of METRA. We note that METRA was specifically designed for high-dimensional pixel-based settings; hence, its weaker performance in URLB is consistent with this intended scope. Our work focuses on isolating and validating the effect of balancing exploration and diversity during skill pretraining on standard control environments, but extending AMPED to pixel-based domains is an important direction for future work, and we appreciate the reviewer highlighting this connection.
>
> ### 2. Efficiency Analysis
>
> To address the reviewer’s request for efficiency comparisons, we added return-time series plots across four evaluation metrics for Quadruped in Section N.1, and will include the corresponding plots for Walker and Jaco before the rebuttal period ends (these are currently running due to computational cost). These plots show that AMPED consistently surpasses the baselines throughout downstream fine-tuning. Additionally, Appendix I.7 reports wall-clock comparisons, demonstrating that AMPED achieves strong empirical performance while maintaining reasonable computational cost.
>
> ### 3. Clarifying Our Contributions
> We appreciate the reviewer’s question regarding methodological novelty. While AMPED combines several existing components, our contributions go beyond a simple aggregation:
>
> - **Balancing exploration and diversity:** We provide empirical evidence that jointly handling both objectives is critical in SBRL, a dimension underexplored in prior work.
> - **Gradient-conflict perspective:** We identify and measure conflict between exploration and diversity objectives, and adapt PCGrad with a probabilistic projection mechanism tailored to this setting.
> - **Exploration design:** We show that combining RND with a particle-based entropy estimator, though not previously explored in skill-based RL, significantly strengthens exploration.
> - **MI estimation for skills:** We introduce AnInfoNCE as an MI estimator for skill diversification. Although AnInfoNCE is an existing method, it has not been applied to skill-based RL before, and our ablations demonstrate clear benefits.
> - **Theory:** We provide a theoretical analysis highlighting the role of diversity in skill-based RL, an aspect less theoretically grounded than exploration in prior literature.
>
> Together, these components offer a coherent framework that consistently improves skill-based RL performance across tasks.
>
> We again thank the reviewer for the insightful feedback, which has allowed us to broaden our empirical analysis. We hope our responses satisfactorily resolve the concerns raised and kindly ask the reviewer to consider them in their final evaluation.

---

> > ### Comment · Reviewer_3YZa · 2025-11-27
> >
> > Thank you for your response. I have raised my score.

---

> > > ### Author Response · Authors · 2025-11-27
> > >
> > > Thank you for the careful evaluation and valuable comments. We appreciate the time you invested in reviewing our work and the constructive insights that guided several improvements.

---

### Official Review · Reviewer_1a1c · 2025-11-03

**Soundness:** 4
**Presentation:** 4
**Contribution:** 3
**Rating:** 4
**Confidence:** 4

**Summary:**

The authors propose AMPED, a skill-based reinforcement learning framework that balances exploration and skill diversity objectives during pretraining. The method treats these as competing objectives in a multi-objective RL setting and applies gradient surgery (PCGrad) to resolve conflicts between diversity gradients (from AnInfoNCE) and explroation gradients (from particle-based entropy and RND). Those skills are exploited in the following fine-tuning phase, in which a SAC-based skill selector adaptively chooses skills for downstream tasks.

**Strengths:**

- The paper is very clear and easy to follow, and the proposed method is well-motivated.
- Strong results on several tasks from the URLB benchmark with comparisons against 7 baselines, with consistent improvements.
- Table 2 effectively demonstrates that each component (RND, AnInfoNCE, gradient surgery, skill selector) contributes meaningfully to performance.
- Figure 5 (Tree Maze) provides intuitive evidence that AMPED achieves both skill separation and state coverage where baselines fail at one or both.

**Weaknesses:**

Section 4.4 claims are problematic in multiple ways:
- Line 200 claims "empirical validation appears in Section 4.4," but Theorem 1 assumes sampling trajectories from the **optimal** policy (requiring access to it), while Section 4.4 measures convergence speed to the optimal policy during fine-tuning. Theorem 1 establishes that diversity reduces sample complexity for skill identification given optimal trajectories—not that diversity accelerates convergence during learning.
- The statement "Combined with higher final returns, this indicates faster convergence of AMPED" is inaccurate, as other variants finish converging before (e.g. BeCL actually converges 90% faster on Flip, and many other comparisons are not statistically significant). The paper should also report the **rate of improvement** (reward gained per step) rather than steps-to-threshold, especially when final returns differ across methods.
- The logic "AMPED has higher diversity than CIC/BeCL" + "AMPED improves faster" $\Rightarrow$ "diversity causes faster improvement" seems flawed. Other algorithmic differences (objective functions, gradient surgery, skill selector) could explain the improvement rate. To isolate diversity's effect, ablations should vary diversity within AMPED itself (e.g., different numbers of skills or projection ratios $p$), not compare across different methods.
- Section 4.4 only evaluates Walker tasks, making it difficult to establish a general relationship between diversity and rate of improvement.

**Questions:**

Table 10 shows skill dimension is fixed at 16 for all URLB experiments. How do diversity and performance vary with different numbers of skills (e.g., 4, 8, 32)? This would help validate whether increased diversity consistently improves results.

---

> ### Author Response · Authors · 2025-11-25
>
> We sincerely thank the reviewer for the thoughtful and constructive feedback. Your comments helped us significantly clarify the theoretical scope of Section 4.4 and refine the empirical analysis. We address each concern in detail below.
>
> ### 1. Relationship between Theorem 1 and Section 4.4
> We agree that the previous version of Section 4.4 did not provide an empirical counterpart that aligns well with Theorem 1. In practice, isolating the effect of diversity is challenging because most interventions that modify diversity simultaneously influence other factors, such as exploration or state coverage. Therefore, rather than attempting to empirically validate the theorem in Section 4.4, we removed that section and instead included a controlled analysis of the number of skills in Section 4.3.
>
> ### 2. Number of Skills
> This ablation demonstrates that diversity does not increase monotonically with skill count and that performance suffers when the skill set is either too small or too large. Intuitively, a small skill set may already provide sufficient coverage of the environment, leaving little opportunity for additional skills to contribute further diversity. Conversely, even in larger environments, diversity is optimized through an approximate mutual-information objective rather than an exact estimator, so increasing the number of skills does not guarantee proportional gains in separation. These observations highlight that selecting an appropriate skill dimension is essential for maximizing diversity and achieving strong downstream performance. Across all three domains, 16 skills (our default) yield the most stable and competitive results.
>
> ### 3. Additional analyses
> We additionally include a return-time series plot for Quadruped in Section N.1 (instead of a rate of improvement graph), and we will provide plots for Walker and Jaco before the end of the rebuttal period (these are still running due to computational cost). These plots show that AMPED consistently surpasses the baselines throughout the downstream fine-tuning process.
>
> We are grateful for the reviewer’s comments, which significantly improved both the theoretical discussion and experimental scope of the paper. We believe our clarifications address the raised questions, and we would appreciate the reviewer’s consideration of these responses in their overall judgement.

---

> > ### Comment · Reviewer_1a1c · 2025-11-27
> >
> > Thank you very much for your reply and for the details. All my concerns have been addressed, and I have increased my score.

---

> > > ### Author Response · Authors · 2025-11-27
> > >
> > > Thank you very much for the thoughtful follow-up and the updated score. We sincerely appreciate your careful reading and constructive feedback throughout the review process.

---

### Meta-Review · Area_Chair_GzKa · 2025-12-25

**Summary:**

The four reviewers agree that the paper is clearly written, empirically strong, and addresses an important practical issue in skill-based RL: jointly handling exploration and skill diversity. The proposed AMPED framework combines particle-based entropy + RND for exploration, AnInfoNCE for skill diversity, and a gradient-surgery mechanism plus a skill selector, and it achieves consistent gains on URLB and maze benchmarks with solid ablations. The main concerns raised were: (1) the strength and scope of the theoretical claim connecting diversity to fine-tuning sample complexity, and how well it is reflected empirically; (2) the degree of methodological novelty, since many components (PCGrad, RND, InfoNCE/AnInfoNCE, hierarchical skill selection) are existing ideas; (3) the limited benchmark diversity (no pixel-based / more complex environments) and algorithmic complexity/efficiency; and (4) several clarity issues in the theory, metrics, and description of the skill selector. The rebuttal and revision directly address most technical and clarity concerns, and all reviewers who responded raised their scores, leading to a positive overall consensus in favor of acceptance.

**Reviewer Concerns:**

Most substantive reviewer concerns were largely addressed in the rebuttal and revision. In particular, the authors successfully clarified the scope of the theory, improved the presentation of key definitions and metrics, added missing ablations (such as the number of skills), included additional baselines and efficiency analyses, and removed or softened claims that previously appeared overstated. These changes resolved the main issues regarding clarity, empirical support for claims, and the alignment between the theoretical discussion and the experiments.

Some concerns, however, remain partially outstanding. The theoretical result still operates under simplified assumptions and does not fully close the gap to the practical learning dynamics observed in the full algorithm, and the overall methodological novelty may reasonably be viewed as incremental, given that the approach builds on several existing components. In addition, the experimental setting remains focused on standard control benchmarks rather than more complex or pixel-based domains. These limitations do not undermine the main contributions of the paper, but they do slightly constrain its generality and perceived scope.

**Reviewer Scores:**

•	Reviewer 1a1c: explicitly states that all concerns have been addressed and that they increased their score.

•	Reviewer 3YZa: explicitly states “I have raised my score.”


•	Reviewer 8RFz: no explicit numerical update, but their technical clarification questions (theorem interpretation, metrics, gradient curves) have been addressed in the revision.

•	Reviewer B5oH (mrH4): explicitly states that the paper was already in good shape, that concerns were minor, and that they increased their score to 8.

Overall, after rebuttal and discussion, there is a strong consensus shift toward acceptance: two initially borderline-negative reviewers moved into the accept region, one reviewer remains positive, and one moved to a clear-accept score. I recommend acceptance.

---

### Decision · Program_Chairs · 2026-01-26

Accept (Poster)